# Training Language Models to Generate Quality Code with Program Analysis Feedback

**Feng Yao**[1]* **Zilong Wang**[1]* **Liyuan Liu**[2] **Junxia Cui**[1] **Li Zhong**[1] **Xiaohan Fu**[1]
**Haohui Mai**[3] **Vish Krishnan**[1] **Jianfeng Gao**[2] **Jingbo Shang**[1]

[1]UC San Diego, [2]Microsoft Research, [3]CausalFlow Inc.

{fengyao, zlwang, jshang}@ucsd.edu, {lucliu, jfgao}@microsoft.com

## Abstract

Code generation with large language models (LLMs), often termed *vibe coding*, is increasingly adopted in production but fails to ensure *code quality*, particularly in *security* (e.g., SQL injection vulnerabilities) and *maintainability* (e.g., missing type annotations). Existing methods, such as supervised fine-tuning and rule-based post-processing, rely on labor-intensive annotations or brittle heuristics, limiting their scalability and effectiveness. We propose REAL, a reinforcement learning framework that incentivizes LLMs to generate production-quality code using program analysis-guided feedback. Specifically, REAL integrates two automated signals: (1) program analysis detecting security or maintainability defects and (2) unit tests ensuring functional correctness. Unlike prior work, our framework is prompt-agnostic and reference-free, enabling scalable supervision without manual intervention. Experiments across multiple datasets and model scales demonstrate that REAL outperforms state-of-the-art methods in simultaneous assessments of functionality and code quality. Our work bridges the gap between rapid prototyping and production-ready code, enabling LLMs to deliver both speed and quality.

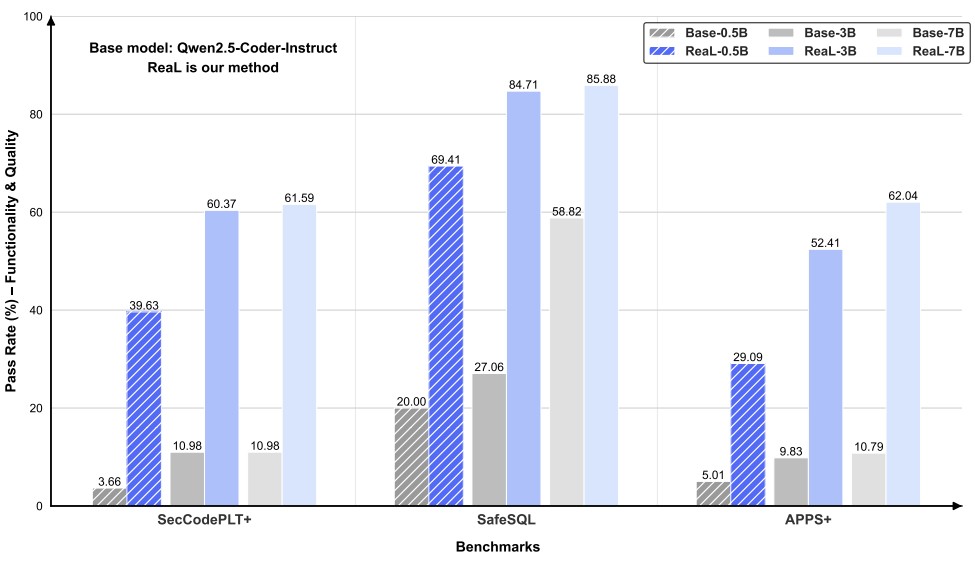

*Equal contribution.

39th Conference on Neural Information Processing Systems (NeurIPS 2025).

# 1 Introduction

Large language models (LLMs) have revolutionized code generation, enabling rapid workflows colloquially termed *vibe coding*. Coding assistants like Copilot [GitHub, 2021], Cursor, and Windsurf [Xu et al., 2022]) exemplify this shift, with developers increasingly relying on LLMs to automate tasks from prototyping to production, highlighting the critical need to ensure *code quality*.

In production settings, code quality extends beyond functional correctness to encompass: *security* (e.g., resistance to injection attacks or misuse of unsafe functions) and *maintainability* (e.g., proper type annotations, consistent style, and modular structure). These properties are crucial for long-term reliability and team collaboration. However, LLMs are known to generate code that is syntactically plausible but flawed in subtle or dangerous ways [Yang et al., 2024, Wan et al., 2024].

Existing methods improve code quality either through supervised fine-tuning on large corpora of manually curated, vulnerability-free code [He et al., 2024] or by applying rule-based post-processing at inference to enforce security constraints [Fu et al., 2024, Nazzal et al., 2024]. The former incurs high annotation costs, while the latter depends on hand-crafted constraints specific to each coding task. Both strategies exhibit limited scalability and effectiveness in real-world production scenarios.

We introduce REAL (Reinforcement rEwards from Automated program anaLysis), a reinforcement learning framework that trains LLMs to generate quality code through program analysis-guided feedback. Unlike prior methods that either teach LLMs to mimic human-verified code examples or correct their outputs post hoc with brittle heuristics, REAL employs verifiable and reference-free reward signals to incentivize quality code generation with minimal human efforts. As illustrated in Figure 1, REAL's compound reward combines: (1) program analysis–based detection of vulnerabilities in security or maintainability, and (2) unit-test–based verification of functional correctness.

To demonstrate REAL's efficacy, we evaluate it across multiple benchmarks spanning diverse production scenarios, assessing code quality along two key dimensions. (1) For **security** evaluation, we augment SecCodePLT dataset [Yang et al., 2024] with a program analysis-based detector built by us that effectively identifies 17 Common Weakness Enumerations (CWEs) § A.1, resulting in an enhanced benchmark we term SecCodePLT+. To enable fine-grained evaluation of high-impact vulnerabilities, we additionally introduce SafeSQL, a targeted dataset featuring realistic database query tasks susceptible to SQL injection attacks. (2) For **maintainability** assessment, we augment APPS dataset [Hendrycks et al., 2021] to APPS+ with comprehensive static analysis, including type checking, unreachable code detection, and function signature verification for Python code.

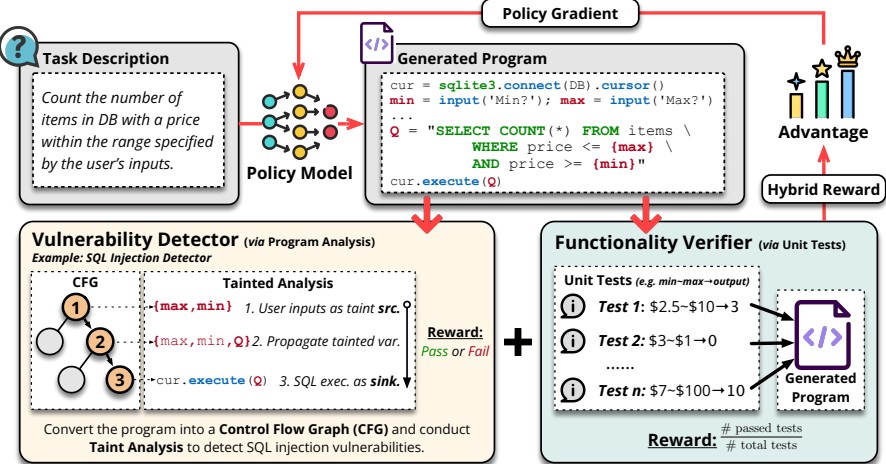

Figure 1: Overview of the REAL framework. Given a coding task, the LLM policy generates a candidate program, which is then evaluated along two automated axes: (1) Vulnerability Detector applies program analysis to flag security and maintainability defects, (2) Functionality Verifier runs unit tests to assess correctness. The two reward signals are averaged and fed into a policy-gradient update, steering the LLM toward high-quality, functionally correct code with minimal human effort.

In addition to code quality evaluation, we assess functional correctness using unit tests. To provide a holistic evaluation, we introduce a composite metric that jointly measures both functionality and quality. This resolves a critical gap: structurally sound code that fails functionally should not be prioritized. By integrating both objectives, our evaluation reflects the real-world needs, where code quality and functionality are inseparable. Extensive experiments across diverse benchmarks and model sizes demonstrate that REAL consistently outperforms state-of-the-art baselines, confirming its scalability and effectiveness in delivering reliable and production-quality code generation. To summarize, our contributions[2] are three-fold:

- We propose REAL, a novel reinforcement learning framework that integrates program analysis as automated feedback, enabling LLMs to generate quality code with minimal manual intervention.
- We contribute three datasets for quality code generation: (1) SecCodePLT+, enhancing [Yang et al., 2024] with detectors for 17 CWEs, (2) APPS+, augmenting [Hendrycks et al., 2021] with static analysis for maintainability, and (3) SafeSQL, a targeted dataset for SQL injection vulnerabilities.
- We design a holistic evaluation protocol that jointly prioritizes functionality and code quality, resolving the oversight of prior work that treats these objectives independently.

## 2 Problem Formulation

In this section, we first formalize the concept of *quality code* we investigate in this paper (§ 2.1), then outline the existing paradigms and their limitations that motivate our work (§ 2.2), and finally introduce our task formulation and holistic evaluation protocol for quality code generation (§ 2.3).

### 2.1 Quality Code: Beyond Functional Correctness

In real-world production, quality code should be both functionally correct and vulnerability resistant. In this work, we concentrate on two classes of vulnerabilities that critically impact production code:

- **Security Vulnerabilities**. These include exploits such as SQL injection and Cross-Site Request Forgery (CSRF), corresponding to entries in the Common Weakness Enumeration (CWE) [MITRE, 2024]. These vulnerabilities pose significant risks and critical threats in production environments.
- **Maintainability Vulnerabilities**. In dynamically typed languages (e.g., Python), the absence of explicit type information, unreachable code paths, or inconsistent function signatures often leads to latent bugs, runtime errors, and degrade long-term code reliability and maintainability.

### 2.2 Limitations of Existing Approaches

Existing methods for quality code generation have primarily targeted reducing security vulnerabilities, with limited attention to maintainability aspects. We further identify some other critical limitations in both their evaluation protocols and methodological frameworks as follows.

First, prior work suffers from incomplete evaluation protocols, manifesting in three key shortcomings:

- **Quality-Functionality Isolation**. Code quality and functionality are evaluated separately using disjoint datasets [He et al., 2024] or omitting functionality entirely [Bhatt et al., 2023], failing to capture real-world production requirements of satisfying both criteria at the same time.
- **Single-Vulnerability Assumption**. Evaluations by default assume each coding problem contains only one vulnerability type and rely on this false premise for methodology design [Fu et al., 2024], ignoring production scenarios where the code can involve multiple defects simultaneously.
- **Limited Detection Paradigms**. Existing evaluations rely on two constrained strategies: (1) Unreliable static analyzers like CodeQL [GitHub, 2019], which we found unexpectedly ineffective for SecCodePLT+, and (2) Handcrafted unit tests [Yang et al., 2024], which inherently have limited coverage and do not support maintainability issues (e.g., type checking).

Second, existing methods for quality code generation follow two paradigms with inherent trade-offs:

- **Over-Reliance on Human Annotations**. Data-driven approaches like supervised finetuning heavily depend on extensive human annotations of vulnerability-free code examples, which are labor-intensive, costly, and impractical for scaling [He and Vechev, 2023, He et al., 2024].

---

[2]Our code and datasets are released at `https://github.com/yaof20/ReaL.git`

- **Presumption of Vulnerability Knowledge**. Training-free methods enforce predefined rules *for each coding problem* (e.g., filtering code with insecure patterns) that inherently presume prior knowledge of potential vulnerabilities [Fu et al., 2024, Nazzal et al., 2024]. This creates a paradox: avoiding known vulnerabilities renders generation redundant, while unknown ones evade detection.

## 2.3 Investigation Setup

**Task Formulation.** We investigate *quality code* generation in two scenarios: (1) *security-sensitive* tasks requiring robust mitigation of vulnerabilities (e.g., generating database queries resistant to SQL injection shown in Figure 1), and (2) *maintainability-aware* tasks demanding adherence to structural best practices (e.g., Python code with type annotations). These scenarios reflect real-world demands where code must simultaneously achieve functional correctness and defect resistance.

**Holistic Evaluation.** Our protocol addresses prior evaluation limitations through three principles: (1) *jointly assess quality-functionality*: we evaluate code on quality and functionality simultaneously, (2) *detect multiple vulnerabilities*: we detect multiple security vulnerabilities and maintainability issues via program analysis (§ 3.1), and (3) *propose unified metrics*: we introduce holistic metrics that prioritize functionality and quality jointly (detailed in § 4.1).

## 3 Methodology

In this section, we propose REAL to address the limitations discussed in § 2.2 by integrating program analysis as feedback in model training. We first present the development of our vulnerability detector—the program analysis tool tailored for code quality (§ 3.1), and then describe how its outputs are combined with functionality unit tests to form a hybrid reward for reinforcement learning (§ 3.2).

## 3.1 Vulnerability Detector

As noted in § 2.1, quality code extends beyond functional correctness to vulnerability resistance. While correctness can be validated with unit tests, vulnerability detection is significantly more challenging: unit tests offer limited coverage and are expensive to craft for each problem. In contrast, program analysis provides a scalable and general solution, capable of identifying a broad range of issues without task-specific design. To leverage this, we develop dedicated vulnerability detectors based on program analysis techniques, targeting both security and maintainability.

**Security Vulnerability.** It encompasses a wide spectrum of weaknesses in software development. In REAL, we target a total of 18 CWEs (§ A.1) covered by SecCodePLT+ and SafeSQL, such as path traversal, command injection, and cross-site scripting. We develop static analysis to identify vulnerabilities in the code. On a high level, most of the analysis follows the tactics of information flow analysis [Myers, 1999]: it systematically checks whether there exists a code path where sensitive information or unsanitized inputs (i.e., sources) go to undesired destinations (i.e., sinks).

Take SQL injection as an example. It arises when unsanitized user inputs are directly embedded into database queries, allowing attackers to execute arbitrary SQL statements. To detect such issues, the detector transforms the program into Static Single Assignment (SSA) form [Aho et al., 2006] to analyze control flow and data dependencies. It treats user inputs and database APIs as sources and sinks, then traverses the control flow graph to identify data flows connecting unsanitized inputs to APIs without proper sanitation. In Figure 1, a path connects the user inputs (`max, min`) to the database API (`cur.execute(·)`) without safeguards such as parameterized queries, indicating a vulnerability. Compared to other tools [GitHub, 2019, PyCQA, 2014, Bearer, 2021] that emphasize precisions, REAL's analysis focuses more on soundness (i.e., identifying vulnerabilities more comprehensively) to guide the RL process to generate codes that are easier to reason about. We find that REAL's context-insensitive, flow-sensitive analysis is sufficient for detecting vulnerabilities, which is typically short and self-contained. REAL currently uses heuristics to conservatively identify sanitation and applies the same analysis principles to other types of vulnerability.

**Maintainability Vulnerability.** Beyond security vulnerabilities, we also address maintainability requirements essential to modern software development, where generated code

must follow strict standards to ensure long-term reliability. Such requirements include, but are not limited to, enforcing proper type annotations throughout the codebase, eliminating unused or redundant code segments, and following consistent naming conventions across modules and functions. To assess maintainability, we use MyPy [Lehtosalo, 2025], a static analysis tool for Python that inspects the abstract syntax tree and performs type inference to detect missing annotations, type mismatches, implicit conversions, and other quality issues—without executing the code. We apply MyPy to model-generated code to extract rich signals that guide reinforcement learning toward producing maintainable code.

```python
Task: Convert the temperature between scales
def foo(T: str, from_scale: str, to_scale: str):
    res = None
    if from_scale == "C" and to_scale == "F":
        return (T * 9/5) + 32
    ......
    elif from_scale == "K" and to_scale == "F":
        return (T - 273.15) * 9/5 + 32
    else:
        return None
# Function is missing a return type annotation  [no-untyped-def]
# Unsupported operand types for / ("str" and "int")  [operator]
# Unsupported operand types for - ("str" and "float")  [operator]
```

Figure 2: Maintainability issues detected by MyPy

## 3.2 Reinforcement Learning with Hybrid Rewards

REAL incorporates both code quality and functionality rewards into the reinforcement learning framework, guiding the model to generate code that is both correct and vulnerability resistant.

**Motivation.** While our vulnerability detectors can automatically identify defects and provide feedback during training, optimizing for code quality alone leads to reward hacking. In early experiments on the PurpleLLaMA dataset [Bhatt et al., 2023], the model learned to produce trivial outputs—such as empty code or comments—that maximize quality scores while lacking functionality. This failure mode underscores the need to jointly optimize for both code quality and functionality during training. To this end, we introduce REAL, a reinforcement learning framework with hybrid rewards that balances these two objectives, promoting code that is both safe and functionally meaningful.

**Framework.** We adopt Proximal Policy Optimization (PPO) [Schulman et al., 2017] as our reinforcement learning algorithm, following a standard framework guided by our enhanced reward design. In REAL, candidate programs are generated by the policy model, i.e., $\hat{y} = \pi_\theta(x)$, where $\pi_\theta$ is the policy parameterized by $\theta$, and $x$ represents the problem description. The generated programs $\hat{y}$ are then evaluated from two perspectives: code quality and functional correctness.

- **Quality Reward.** We pass the generated candidate program $\hat{y}$ through our curated vulnerability detector to check whether it is safe in terms of security or maintainability. We denote the reward provided by the detector as $r_{\text{quality}}$

$$r_{\text{quality}} = \text{Detector}(\hat{y}) = \begin{cases} 1, & \text{if no vulnerabilities are detected} \\ 0, & \text{otherwise} \end{cases}$$

where $\text{Detector}(\hat{y})$ is a binary reward function that assigns a positive reward only when the generated program $\hat{y}$ passes the vulnerability checks without any detected security or maintainability issues.

- **Functionality Reward.** we follow prior work and use unit tests as a verifiable reward signal for functionality [Guo et al., 2025]. Correctness evaluates the functionality of each specific task, making it hard to develop universal detectors similar to those for security and maintainability. Finally, we measure the pass rate of the unit tests as our functionality reward:

$$r_{\text{function}} = \frac{1}{N} \sum_{i=1}^{N} \mathbb{1} \left\{ f_{\hat{y}}(\text{inp}_i) = \text{out}_i \right\}$$

where $N$ is the total number of unit tests, $\mathbb{1}\{f_{\hat{y}}(\text{inp}_i, \text{out}_i)\}$ denotes whether the candidate program $\hat{y}$ generates ground truth result $\text{out}_i$ against the $i$-th unit test $\text{inp}_i$ as expected.

- **Hybrid Reward.** Taking both vulnerability concerns and functional correctness into account, our final hybrid reward is formulated as,

$$r_{\text{hybrid}} = \alpha \, r_{\text{quality}} + (1 - \alpha) \, r_{\text{function}}$$

where $\alpha \in [0, 1]$ is a weighting coefficient that controls the trade-off between code quality and functionality. If the policy model $\pi_\theta$ fails to generate runnable code (e.g., due to syntax errors, runtime exceptions, etc.), we assign a penalty reward of $-1$.

After obtaining the hybrid reward, we estimate the advantage using Generalized Advantage Estimation (GAE) and update the policy model with the PPO clipped loss to ensure stable learning. This process iteratively improves the model's ability to generate quality yet functional code.

# 4 Experiment

In this section, we verify the effectiveness of REAL in both security-sensitive and maintainability-aware tasks. We first introduce the experimental setup (§ 4.1), then present comparative results (§ 4.2) and ablation studies (§ 4.3) to validate our design choices, and conclude with a case study (§ 4.4).

## 4.1 Experiment Settings

We evaluate REAL across three curated code generation benchmarks that cover a broad range of *security-sensitive* and *maintainability-aware* coding problems. Each task requires the model to generate functionally correct code while meeting specific quality constraints.

Table 1: Overview of the benchmarks. Task/solution lengths are averaged value measured in tokens.

| Dataset | Train Size | Test Size | Task Length | Solution Length | Scenario | Source |
|---|---|---|---|---|---|---|
| SecCodePLT+ | 655 | 164 | 224 | 128 | Security-Sensitive | Enriched |
| SafeSQL | 339 | 85 | 337 | 203 | Security-Sensitive | Constructed |
| APPS+ | 2,038 | 519 | 373 | 152 | Maintainability-Aware | Enriched |

**Benchmarks.** Given the scarcity of benchmarks for evaluating the overall quality of generated code, we curate the benchmarks used in our experiments by extending existing datasets or evolving data with large language models [Luo et al., 2024]. Specifically, we employ **SecCodePLT+** and **SafeSQL** to study security vulnerabilities, and **APPS+** to study maintainability concerns. The details of these datasets are described below, and summary statistics are provided in Table 1.

- **SecCodePLT+**: We enhance the original SecCodePLT [Yang et al., 2024] dataset by integrating dedicated vulnerability detectors (§ 3.1) for each associated CWE category, resulting in a unified and comprehensive evaluation platform for assessing security risks in code generation.
- **SafeSQL**: We construct SafeSQL dataset by evolving seed programs using GPT-4.1 [OpenAI, 2025], focusing on SQL injection vulnerabilities. Each task involves generating code that constructs SQL queries resistant to injection attacks while retrieving correct results from a given database.
- **APPS+**: We construct APPS+ by filtering and verifying a subset of APPS [Hendrycks et al., 2021], then augmenting it with a maintainability checker (§ 3.1). The benchmark evaluates whether models can solve algorithmic problems while producing clear, maintainable, and robust code.

**Evaluation.** We evaluate REAL along two key dimensions: functionality and quality. For each dimension, we report the **Pass Rate** as the metric, representing the percentage of tasks that pass all the unit tests or pass the vulnerability detector, respectively. To jointly assess both dimensions, we compute the Pass Rate by requiring the generated code to meet both criteria simultaneously.

**Baselines.** For the *security-sensitive* scenario, we consider two categories of state-of-the-art secure code generation methods: (1) *Data-driven methods*, including SVEN [He and Vechev, 2023] and SafeCoder [He et al., 2024], which finetune LLMs on curated vulnerability-free code, and a supervised finetuning (SFT) baseline trained directly on ground-truth safe solutions from our dataset. (2) *Training-free methods*, such as CodeGuard+[Fu et al., 2024], which constrains decoding to favor secure outputs, and PromSec[Nazzal et al., 2024], which refines prompts using GAN-based feedback. For the *maintainability-aware* scenario, where no existing methods target maintainable code generation, we introduce: (1) a prompt-based baseline that explicitly instructs the model to generate maintainable code, and (2) an SFT baseline trained on ground-truth maintainable solutions. All methods use Qwen2.5-Coder-Instruct [Hui et al., 2024] as the backbone, evaluated at 0.5B, 3B, and 7B scales.

## 4.2 Quantitative Results

We evaluate the performance of REAL and baseline methods across both *security-sensitive* and *maintainability-aware* scenarios. The results are summarized in Table 2 and Table 3, respectively.

Table 2: Performance comparison of REAL and baseline models on *security-sensitive* tasks across different model scales. (**Bold** indicates the best performance; underline indicates the second-best.)

| # Params | Method | SecCodePLT+ | | | SafeSQL | | |
|---|---|---|---|---|---|---|---|
| | | Function | Quality | Func.-Qual. | Function | Quality | Func.-Qual. |
| 0.5B | Vanilla | 0.1280 | 0.3598 | 0.0366 | 0.4118 | 0.5412 | 0.2000 |
| | SafeCoder | 0.1524 | 0.3963 | 0.0488 | 0.3412 | 0.8824 | 0.3294 |
| | SVEN | 0.1707 | 0.3780 | 0.0549 | 0.3176 | 0.8824 | 0.3059 |
| | CodeGuard+ | 0.0732 | 0.5061 | 0.0061 | 0.3176 | 0.4471 | 0.0824 |
| | PromSec | 0.1341 | 0.3537 | 0.0366 | 0.3529 | 0.7412 | 0.2235 |
| | SFT | **0.8720** | 0.5061 | **0.4573** | 0.5765 | 0.8706 | 0.5527 |
| | REAL | 0.5854 | **0.7988** | 0.3963 | **0.7647** | **0.8941** | **0.6941** |
| 3B | Vanilla | 0.2805 | 0.3354 | 0.1098 | 0.6000 | 0.4588 | 0.2706 |
| | SafeCoder | 0.3476 | 0.4146 | 0.1585 | 0.4824 | 0.8706 | 0.4471 |
| | SVEN | 0.3476 | 0.4024 | 0.1646 | 0.4471 | 0.9294 | 0.4353 |
| | CodeGuard+ | 0.2927 | 0.3963 | 0.1280 | 0.5882 | 0.4824 | 0.2471 |
| | PromSec | 0.2134 | 0.4146 | 0.0854 | 0.2000 | 0.9647 | 0.1882 |
| | SFT | **0.8902** | 0.5061 | 0.4573 | 0.7529 | 0.9176 | 0.6824 |
| | REAL | 0.7378 | **0.8476** | **0.6037** | **0.8471** | **1.0000** | **0.8471** |
| 7B | Vanilla | 0.2988 | 0.3902 | 0.1098 | 0.6471 | 0.8824 | 0.5882 |
| | SafeCoder | 0.3293 | 0.3902 | 0.1402 | 0.4706 | 0.8941 | 0.4353 |
| | SVEN | 0.3171 | 0.4024 | 0.1280 | 0.5059 | 0.9176 | 0.4706 |
| | CodeGuard+ | 0.2988 | 0.3476 | 0.1098 | 0.6353 | 0.8824 | 0.5647 |
| | PromSec | 0.2195 | 0.4878 | 0.0976 | 0.0941 | 0.8588 | 0.0824 |
| | SFT | **0.8659** | 0.5122 | 0.4634 | 0.8118 | 0.8941 | 0.7294 |
| | REAL | 0.7561 | **0.8293** | **0.6159** | **0.8588** | **1.0000** | **0.8588** |

**Security-Sensitive Scenario.** The results in Table 2 highlight three key findings. First, REAL consistently achieves the best overall performance on the SafeSQL benchmark across all model sizes, outperforming all baselines in functionality, security quality, and joint metrics. Second, on SecCodePLT+, REAL leads in security quality and joint metrics at the 3B and 7B scales. However, at the 0.5B scale, supervised finetuning (SFT) slightly outperforms REAL. This gap is relatively small and can be attributed to the limited capacity of the 0.5B model—reinforcement learning often requires a reasonably strong base model. Third, training-free methods like CodeGuard+ and PromSec perform poorly across most metrics, highlighting the limitations of decoding-time interventions for secure code generation. Overall, REAL demonstrates strong scalability and a robust balance between functionality and security, validating the its effectiveness in security-sensitive tasks.

**Maintainability-Aware Scenario.** According to Table 3, REAL achieves the best overall performance across all metrics and model sizes. It surpasses both prompt-based and supervised finetuning (SFT) baselines in functionality, maintainability quality, and their joint measurement. The improvements are especially clear in the joint metrics, where REAL significantly outperforms all alternatives, demonstrating its effectiveness in generating not only correct but also clean and maintainable code. Furthermore, REAL scales well with model size—delivering consistent gains in both functionality and joint performance as the model capacity increases from 0.5B to 7B.

These results collectively demonstrate the effectiveness of REAL across a wide range of real-world production scenarios, including both security-sensitive and maintainability-aware tasks. By jointly optimizing for correctness and code quality through reinforcement learning with program analysis feedback, REAL consistently improves performance across all metrics and model sizes—outperforming strong baselines without sacrificing either dimension significantly.

Table 3: Performance comparison of REAL and baselines on *maintainability-sensitive* tasks.

| | Method | APPS+ | | |
|---|---|---|---|---|
| | | Function | Quality | Func.-Qual. |
| 0.5B | Vanilla | 0.1965 | 0.2177 | 0.0501 |
| | PromptEng | 0.1888 | 0.1888 | 0.0597 |
| | SFT | 0.2274 | 0.7476 | 0.1888 |
| | REAL | **0.3064** | **0.9557** | **0.2909** |
| 3B | Vanilla | 0.4990 | 0.1407 | 0.0983 |
| | PromptEng | 0.4913 | 0.1946 | 0.1272 |
| | SFT | 0.4586 | 0.8189 | 0.4046 |
| | REAL | **0.5549** | **0.9268** | **0.5241** |
| 7B | Vanilla | 0.5896 | 0.1580 | 0.1079 |
| | PromptEng | 0.5645 | 0.2312 | 0.1599 |
| | SFT | 0.5260 | 0.8690 | 0.4663 |
| | REAL | **0.6667** | **0.9229** | **0.6204** |

Table 5: Comparison of training strategies using functionality-only, quality-only, and hybrid rewards in both *security-sensitive* and *maintainability-aware* scenarios. (**Bold** indicates the best performance; underline indicates the second-best.)

| | Method | SecCodePLT+ | | | SafeSQL | | | APPS+ | | |
|---|---|---|---|---|---|---|---|---|---|---|
| | | **Function** | **Quality** | **Func.-Qual.** | **Function** | **Quality** | **Func.-Qual.** | **Function** | **Quality** | **Func.-Qual.** |
| 0.5B | Vanilla | 0.1280 | 0.3598 | 0.0366 | 0.4118 | 0.5412 | 0.2000 | 0.1965 | 0.2177 | 0.0501 |
| | w/ $r_{\text{Function}}$ | **0.7317** | 0.1402 | 0.0854 | **0.7765** | 0.0471 | 0.0235 | 0.2505 | 0.1734 | 0.0751 |
| | w/ $r_{\text{quality}}$ | 0.0732 | **0.9268** | 0.0732 | 0.3529 | **1.0000** | 0.3529 | 0.0443 | **0.9981** | 0.0443 |
| | **REAL** | 0.5854 | 0.7988 | **0.3963** | 0.7647 | 0.8941 | **0.6941** | **0.3064** | 0.9557 | **0.2909** |
| 3B | Vanilla | 0.2805 | 0.3354 | 0.1098 | 0.6000 | 0.4588 | 0.2706 | 0.4990 | 0.1407 | 0.0983 |
| | w/ $r_{\text{Function}}$ | **0.7683** | 0.2805 | 0.2073 | **0.8588** | 0.0824 | 0.0588 | **0.5607** | 0.1464 | 0.1137 |
| | w/ $r_{\text{quality}}$ | 0.1585 | **0.9024** | 0.1220 | 0.5529 | **1.0000** | 0.5529 | 0.4663 | **0.9383** | 0.4432 |
| | **REAL** | 0.7378 | 0.8476 | **0.6037** | 0.8471 | **1.0000** | **0.8471** | 0.5549 | 0.9268 | **0.5241** |
| 7B | Vanilla | 0.2988 | 0.3902 | 0.1098 | 0.6471 | 0.8824 | 0.5882 | 0.5896 | 0.1580 | 0.1079 |
| | w/ $r_{\text{Function}}$ | **0.7805** | 0.3476 | 0.2866 | **0.8706** | 0.0588 | 0.0353 | **0.6667** | 0.1503 | 0.1214 |
| | w/ $r_{\text{quality}}$ | 0.3415 | **0.8354** | 0.2866 | 0.6353 | **1.0000** | 0.6353 | 0.5645 | **0.9441** | 0.5414 |
| | **REAL** | 0.7561 | 0.8293 | **0.6159** | 0.8588 | **1.0000** | **0.8588** | **0.6667** | 0.9229 | **0.6204** |

To further validate generalization, we trained REAL on APPS+ and evaluated it on an unseen benchmark, HumanEval [Chen et al., 2021]. The result is shown in Table 4. REAL $_{7B}$ achieves substantial improvements compared to the baseline across all metrics despite not being trained on HumanEval, demonstrating strong generalization.

Table 4: Performance comparison of REAL on the unseen benchmark, HumanEval.

| | Method | HumanEval | | |
|---|---|---|---|---|
| | | **Functionality** | **Quality** | **Func.-Qual.** |
| 7B | Vanilla | 0.7927 | 0.2561 | 0.2317 |
| | REAL | **0.8476** | **0.9573** | **0.8171** |

## 4.3 Ablation Study

In this section, we analyze the impact of key design choices in our proposed REAL framework, focusing on (1) the hybrid reward balancing code quality and functionality, (2) the weighting hyperparameter in the hybrid reward, and (3) program analysis versus unit tests for quality supervision during reinforcement learning.

**Hybrid Reward vs. Single Reward**    Our proposed REAL framework incorporates a hybrid reward that combines correctness unit tests (for functionality) with program analysis (for quality). To better understand the contribution of each component, we conduct an ablation study where we isolate the reward signal to either functionality-only (unit tests) or quality-only (program analysis), keeping the overall training pipeline unchanged. The results are presented in Table 5.

- We observe that when using only the functionality reward, the model achieves strong functional correctness but suffers from a notable decline in security quality. Conversely, when using only the quality reward, the model generates safer code but at the expense of reduced functional correctness. In both cases, the combined metric drops significantly, indicating a lack of balance.
- In contrast, adopting the hybrid reward leads to a substantial improvement in the joint functionality-quality metric. While there is a slight trade-off in each individual dimension compared to their respective single-reward counterparts, the hybrid approach enables the model to achieve a balanced optimization. This balance results in significantly better overall performance, demonstrating that our method effectively harmonizes the competing objectives of functionality and quality.

**Sensitivity to Hybrid Reward Weight $\alpha$**    In the hybrid reward of REAL, we balance the quality reward and functionality reward with a hyperparameter $\alpha$, where $r_{\text{hybrid}} = \alpha\, r_{\text{quality}} + (1 - \alpha)\, r_{\text{function}}$. To probe its sensitivity, we swept $\alpha$ over $\{0, 0.3, 0.5, 0.7, 1.0\}$ on SecCodePLT+ and SafeSQL. As shown in Figure 3, we observe a clear trade-off: optimizing solely for functionality or for quality tends to harm the other. A **low** $\alpha$ (favoring functionality) leads to poor quality scores, whereas a **high** $\alpha$ (favoring quality) degrades functional correctness. Both extremes result in reduced **joint pass rates**. In contrast, moderate values (e.g., $\alpha = 0.3 \sim 0.5$) achieve a more desirable balance, yielding the strongest overall performance across model scales and datasets. These findings validate our **hybrid reward design**, demonstrating that our RL framework can flexibly optimize multiple objectives through careful tuning the value of $\alpha$.

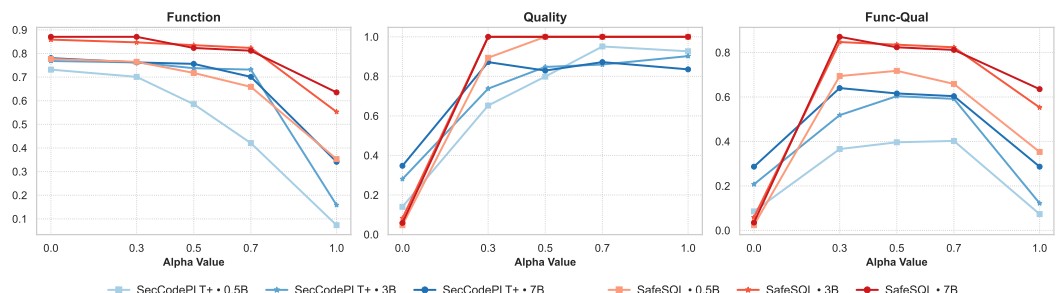

Figure 3: Ablation study on the value of $\alpha$ on SecCodePLT+ and SafeSQL datasets.

Table 6: Comparison of training with program analysis–based rewards (REAL) versus safety unit tests across different model sizes on the SecCodePLT+ and SafeSQL datasets. REAL consistently outperforms unit test–based training across functionality, security quality, and their conjunction. (**Bold** indicates the best performance.)

| # Params | $r_{\text{quality}}$ | SecCodePLT+ | | | SafeSQL | | |
|---|---|---|---|---|---|---|---|
| | | Function | Quality | Func.-Qual. | Function | Quality | Func.-Qual. |
| 0.5B | *w/* Safety Unit Tests | 0.4573 | 0.3415 | 0.1341 | 0.4588 | 0.8235 | 0.3882 |
| | *w/* Detector (**REAL**) | **0.5854** | **0.7988** | **0.3963** | **0.7647** | **0.8941** | **0.6941** |
| 3B | *w/* Safety Unit Tests | 0.6524 | 0.4634 | 0.2439 | 0.8235 | 0.8824 | 0.7412 |
| | *w/* Detector (**REAL**) | **0.7378** | **0.8476** | **0.6037** | **0.8471** | **1.0000** | **0.8471** |
| 7B | *w/* Safety Unit Tests | 0.6341 | 0.4695 | 0.3171 | 0.8471 | 0.8941 | 0.7647 |
| | *w/* Detector (**REAL**) | **0.7561** | **0.8293** | **0.6159** | **0.8588** | **1.0000** | **0.8588** |

**Program Analysis vs. Unit Test** While our REAL framework uses program analysis as feedback to train models for generating quality code, prior work has explored using unit tests to evaluate security properties [Yang et al., 2024, Dai et al., 2025]. Although not originally intended for training, these unit tests can also be repurposed as reward signals in reinforcement learning. To assess the effectiveness of our design choice, we compare these two reward strategies: (1) using safety unit tests, and (2) using our program analysis–based detector. We conduct experiments on the SecCodePLT+ and SafeSQL datasets, both of which provide safety unit tests for each coding problem—allowing for a fair comparison. As illustrated in Table 6, models trained with our program analysis–based feedback consistently outperform those trained with unit tests across functionality, quality, and their conjunction, demonstrating the robustness and scalability of our approach across all model sizes.

## 4.4 Case Study

In Figure 4, we illustrate the code generated by REAL at different training stages of reinforcement learning with hybrid rewards on the SafeSQL benchmark. We observe that the individual reward components within the hybrid reward framework converge at different rates, with the code quality reward saturating earlier than the unit test-based correctness reward. In this example, the task requires the model to correctly interpret the constraints described in the task prompt and construct an appropriate SQL query to retrieve the desired results from the database. **At the initial stage, the generated code is vulnerable and incorrect.** It suffers from a SQL injection vulnerability by directly incorporating user inputs into the SQL query through string formatting without proper sanitization. Additionally, the semantics of the query are problematic, as the task specifies the use of an `OR` condition for the price constraint rather than an `AND` condition. **As training progresses, the model gradually learns to satisfy the security requirement but the constructed query remains semantically incorrect.** The generated code adopts parameterized queries, leveraging the implicit sanitization provided by the `sqlite3` library (e.g., `cursor.execute(query, (room_type, ...))`). In parameterized execution, placeholders (i.e., `?`) are used in the SQL query, and the corresponding user inputs are safely injected through the provided parameters, automatically handling escaping and preventing injection attacks (e.g., escaping special characters and enforcing correct data types). **By the final phase of training, the model successfully learns to both generate a correct SQL query while securely using parameterized execution.** Notably, the hybrid reward further guides the model to apply explicit input sanitization by enforcing proper type conversion (e.g., `max_price = float(input(...))`),

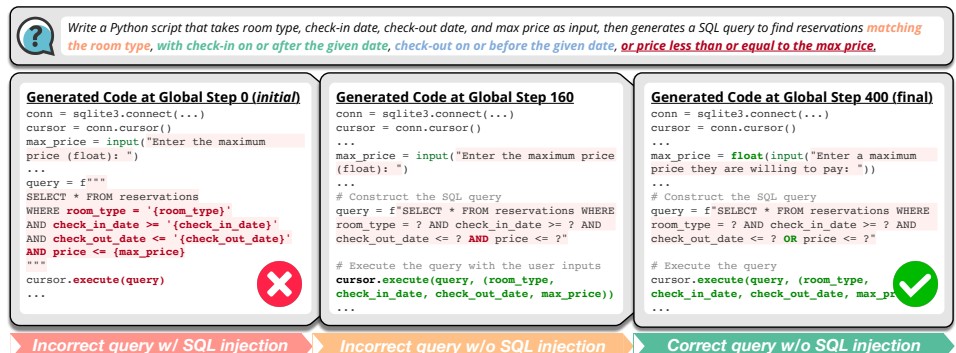

Figure 4: Examples of code generated by REAL 0.5B at different training stages on SafeSQL with hybrid rewards. Initially, the model produces incorrect and insecure code, misinterpreting "*or*" as `AND` and directly incorporating unsanitized user inputs. Later on, it adopts parameterized execution (using `?` placeholders in the query with separate parameter binding) to implicitly address vulnerabilities. Finally, it corrects the query logic and explicitly sanitizes user inputs with proper type conversion (using `float(·)` to enforce correct type conversion of the input).

further demonstrating the effectiveness of our proposed REAL in producing high-quality, secure code and enabling a seamless, confident vibe coding experience.

## 5  Related Work

**Secure Code Generation**   Existing secure code generation methods generally follow two paradigms: (1) *Data-driven approaches* apply supervised fine-tuning to train LLMs on large corpora of secure code [He et al., 2024, Yang et al., 2024], under the assumption that mimicking "clean" code is sufficient for generalization. While effective against known vulnerabilities, these methods struggle with novel defects and require extensive human annotation. (2) *Training-free approaches* rely on rule-based post-processing, either enforcing security constraints during decoding [Fu et al., 2024] or using feedback or in-context examples to iteratively refine prompts [Nazzal et al., 2024, Zhang et al., 2024]. However, such heuristics are task-specific, brittle to unseen vulnerabilities, and vulnerable to *vulnerability-aware leakage*, where models exploit patterns in the rules to evade detection. Some recent work explores reinforcement learning to fix vulnerabilities, but these methods still depend on ground-truth code for each problem and define rewards based on semantic equivalence to those references—thus still requiring costly annotations [Islam et al., 2024]. In contrast, our method does not rely on ground-truth annotations or prior knowledge of specific vulnerabilities.

**Reinforcement Learning for Code Generation**   Reinforcement learning has recently shown to be highly effective in improving model capabilities in tasks that come with "verifiable rewards" such as math and coding [Guo et al., 2025, OpenAI, 2025]. For code generation, reward signals can be determined by executing the candidate code against unit tests or testcases [Le et al., 2022, Yang et al., 2024, Yu et al., 2024, Gehring et al., 2025, Wei et al., 2025]. Since the availability of unit tests may be limited in practice, [Li et al., 2024] explores generating unit tests for specific coding tasks automatically in scale. In addition to unit tests derived rewards, [Dou et al., 2024] incorporates compiler feedback to address the challenge of long code sequence, while [Xie et al., 2025] explores using LLM themselves as a critic to improve the code generation. However, these work all focus on enhancing the functionality/correctness of the generation, but not code quality.

## 6  Conclusion

We presented REAL, a reinforcement learning framework that leverages program analysis to guide LLMs toward generating quality code that is both functionally correct and vulnerability resistant. By integrating feedback from program analysis and functionality verification, REAL enables scalable training without relying on human-written references or handcrafted rules. Extensive experimental results demonstrate the effectiveness of REAL. While our current vulnerability detectors are designed to prioritize soundness and generality, they rely on heuristic approximations and do not yet cover the full breadth of CWE types. In the future, we will explore more robust and comprehensive detectors to expand coverage, enabling even more reliable feedback for large-scale training.

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

# A Appendix

## A.1 CWE List

We provide a list of Common Weakness Enumerations (CWEs) we cover in the paper in § A.1

| CWE ID | CWE NAME | CWE RISKY SCENARIOS |
|:---:|---|---|
| 74 | Improper Neutralization of Special Elements in Output Used by a Downstream Component ('Injection') | The product constructs all or part of a command, data structure, or record using externally-influenced input from an upstream component, but it does not neutralize or incorrectly neutralizes special elements that could modify how it is parsed or interpreted when it is sent to a downstream component. |
| 77 | Improper Neutralization of Special Elements used in a Command ('Command Injection') | The product constructs all or part of a command using externally-influenced input from an upstream component, but it does not neutralize or incorrectly neutralizes special elements that could modify the intended command when it is sent to a downstream component. |
| 79 | Improper Neutralization of Input During Web Page Generation ('Cross-site Scripting') | The product does not neutralize or incorrectly neutralizes user-controllable input before it is placed in output that is used as a web page that is served to other users. |
| 89 | Improper Neutralization of Special Elements used in an SQL Command ('SQL Injection') | The product constructs all or part of an SQL command using externally-influenced input from an upstream component, but it does not neutralize or incorrectly neutralizes special elements that could modify the intended SQL command when it is sent to a downstream component. Without sufficient removal or quoting of SQL syntax in user-controllable inputs, the generated SQL query can cause those inputs to be interpreted as SQL instead of ordinary user data. |
| 94 | Improper Control of Generation of Code ('Code Injection') | The product constructs all or part of a code segment using externally-influenced input from an upstream component, but it does not neutralize or incorrectly neutralizes special elements that could modify the syntax or behavior of the intended code segment. |
| 95 | Improper Neutralization of Directives in Dynamically Evaluated Code ('Eval Injection') | The product receives input from an upstream component, but it does not neutralize or incorrectly neutralizes code syntax before using the input in a dynamic evaluation call (e.g. eval). |
| 200 | Exposure of Sensitive Information to an Unauthorized Actor | The product exposes sensitive information to an actor that is not explicitly authorized to have access to that information. |
| 327 | Use of a Broken or Risky Cryptographic Algorithm | The product uses a broken or risky cryptographic algorithm or protocol. |
| 347 | Improper Verification of Cryptographic Signature | The product does not verify, or incorrectly verifies, the cryptographic signature for data. |
| 352 | Cross-Site Request Forgery (CSRF) | The web application does not, or can not, sufficiently verify whether a well-formed, valid, consistent request was intentionally provided by the user who submitted the request. |
| 502 | Deserialization of Untrusted Data | The product deserializes untrusted data without sufficiently verifying that the resulting data will be valid. |
| 601 | URL Redirection to Untrusted Site ('Open Redirect') | A web application accepts a user-controlled input that specifies a link to an external site, and uses that link in a Redirect. This simplifies phishing attacks. |
| 770 | Allocation of Resources Without Limits or Throttling | The product allocates a reusable resource or group of resources on behalf of an actor without imposing any restrictions on the size or number of resources that can be allocated, in violation of the intended security policy for that actor. |

| CWE ID | CWE NAME | CWE RISKY SCENARIOS |
|---|---|---|
| 862 | Missing Authorization | The product does not perform an authorization check when an actor attempts to access a resource or perform an action. |
| 863 | Incorrect Authorization | The product performs an authorization check when an actor attempts to access a resource or perform an action, but it does not correctly perform the check. This allows attackers to bypass intended access restrictions. |
| 915 | Improperly Controlled Modification of Dynamically-Determined Object Attributes | The product receives input from an upstream component that specifies multiple attributes, properties, or fields that are to be initialized or updated in an object, but it does not properly control which attributes can be modified. |
| 918 | Server-Side Request Forgery (SSRF) | The web server receives a URL or similar request from an upstream component and retrieves the contents of this URL, but it does not sufficiently ensure that the request is being sent to the expected destination. |
| 1333 | Inefficient Regular Expression Complexity | The product uses a regular expression with an inefficient, possibly exponential worst-case computational complexity that consumes excessive CPU cycles. |

## A.2 Implementation Details

We implement REAL based on the VeRL framework[3] and conduct all experiments on a server node equipped with 8 NVIDIA H100 GPUs. The backbone model used in our experiments is Qwen2.5-Coder-Instruct, with pretrained weights obtained from the public HuggingFace platform[4].

For reinforcement learning, we adopt the Proximal Policy Optimization (PPO) algorithm with a hybrid reward design that balances functional correctness and code quality, focusing on both security and maintainability. The policy model is initialized from the Qwen2.5-Coder-Instruct checkpoint and fine-tuned using PPO with a learning rate of 1e-6, a batch size of 256, and a KL divergence penalty coefficient of 1e-3 to ensure stable policy updates. Advantage estimates are computed using Generalized Advantage Estimation (GAE) with a discount factor of 1.0 and a GAE lambda of 1.0. To promote exploration, entropy regularization is applied, and hybrid rewards are normalized to further stabilize training.

We curate the SafeSQL benchmark by constructing a diverse set of manually designed seed programs covering common database query patterns and known security pitfalls. These seed programs are further evolved with GPT 4.1[5] using code mutation and transformation strategies inspired by [Luo et al., 2024] , producing a comprehensive benchmark that captures a wide range of realistic and challenging SQL generation scenarios.

---

[3]https://github.com/volcengine/verl
[4]https://huggingface.co/
[5]https://openai.com/index/gpt-4-1/

