# OpenReview forum: "Training Language Models to Generate Quality Code with Program Analysis Feedback"
_NeurIPS.cc/2025/Conference — NeurIPS 2025 poster_

### Official Review · Reviewer_qzWn · 2025-06-22

**Clarity:** 4
**Significance:** 3
**Originality:** 4
**Rating:** 5
**Confidence:** 3

**Summary:**

The authors propose training an LLM model to generate safe, functional, and maintainable code. To measure safety, they use program analysis-based techniques. For functionality, they use unit test cases. For maintainability, they use some simple static analysis tools.

They augment SecCodePLT by identifying 17 CWEs to create the SecCodePLT+ dataset.
They create a new dataset called SafeSQL, a targeted dataset with realistic database query tasks susceptible to SQL injection attacks.

The authors train the model in an RL setting, with a score that is a tradeoff between security and utility. They can also detect multiple vulnerabilities simultaneously.

The quality reward measures the presence of a vulnerability in the code.
The functionality reward measures the pass rate of the model.
The hybrid reward combines the above metrics using a linear combination of the scores.

**Questions:**

Will the authors open-source the tool they created for the detection of 17+ vulnerabilities?


(Q) Can the authors provide insight into the following observation: Based on the results, the SFT model seems to be a very strong competitor model, especially for the functionality metric on SecCodePLT+.

Can the authors share insight into why this is the case?

**Ethical Concerns:**

["NO or VERY MINOR ethics concerns only"]

**Final Justification:**

I appreciate the authors’ thoughtful and timely rebuttal. It clarified several points from my review and addressed some questions at a high level. Balancing the helpful clarifications against the remaining limitations, my overall evaluation is unchanged, and I maintain my original score.

**Limitations:**

Although there seems to be an in-depth discussion on the limitations of related work, I did not find any explicit paragraph on the limitations of REAL, and also the possible negative societal impacts of their training strategy.

LLMs that have the ability to generate both safe and functional code can have important implications on the software engineering industry through the means of automation.

**Quality:**

3

**Strengths And Weaknesses:**

Strengths

The paper is clear and well written. It is easy to follow and understand. The key contributions are evident in the writing, in both the datasets and in the methodology. The paper proposes a novel technique to integrate both the practical aspects of code safety and functionality.  It addresses an important gap in existing work and bridges this gap through a joint optimization framework that optimizes for both functionality and security.

The approach is novel and is unique. It is an elegant way to combine both the functional and security scores in a unified format. It also seems like an original idea and a good contribution to this downstream task. The problem being tackled is significant and important to this research area.


Weakness

Based on the results, the SFT model seems to be a very strong competitor model, especially for the functionality metric on SecCodePLT+.

---

> ### Author Rebuttal · Authors · 2025-07-31
>
> We sincerely thank Reviewer qzWn for their positive and thoughtful assessment. We appreciate the recognition of our paper’s clarity, the novelty of our joint optimization framework, and the significance of addressing both functionality and security in code generation.
>
> We address the reviewer’s concerns as follows:
>
> **Response to Weakness**
>
> > *“Can the authors provide insight into the following observation: Based on the results, the SFT model seems to be a very strong competitor model, especially for the functionality metric on SecCodePLT+.”*
> >
>
> This is a critical point! Indeed, the SFT model performs strongly on **functionality** in SecCodePLT+, even surpassing REAL. We attribute this to the relatively straightforward nature of the benchmark: SFT benefits from direct supervision using ground-truth solutions, allowing it to learn precise patterns that match test cases.
>
> In contrast, **REAL**, trained via reinforcement learning, **does not rely on ground-truth answers**. Instead, it explores candidate solutions and learns from reward feedback. While this can lead to slightly lower functionality in simple benchmarks, it offers greater flexibility in satisfying **complex and diverse constraints**—such as the 17 CWEs and broader code quality goals. As shown in our results, REAL excels in **security and maintainability**, demonstrating its ability to go beyond pattern-matching and adapt to real-world quality requirements.
>
> In short, while SFT is strong at mimicking known solutions, REAL is designed to generalize and optimize for broader, verifiable objectives without relying on heavy supervision.
>
> ---
>
> **Response to Question**
>
> > *“Will the authors open-source the tool they created for the detection of 17+ vulnerabilities?”*
> >
>
> Yes, we recognize the importance of open-sourcing to support the research community. We plan to release the vulnerability detection tool after completing internal validation and adding documentation to ensure its usability and reliability. This will help promote reproducibility and enable future work on quality code generation.
>
> ---
>
> **Response to Limitation**
>
> > *“Although there seems to be an in-depth discussion on the limitations of related work, I did not find any explicit paragraph on the limitations of REAL, and also the possible negative societal impacts of their training strategy.”*
> >
>
> Thank you for raising this important point. A clear discussion of REAL’s limitations and potential societal impacts will indeed help strengthen the paper.
>
> - **Limitations:** REAL currently focuses on **single-turn code generation**, where the model outputs a complete solution in one step. However, many real-world scenarios involve **multi-turn or interactive coding**, where bugs and vulnerabilities may emerge through complex interactions across multiple code fragments. Detecting and resolving such issues is more challenging and remains an open area for extension of our framework.
> - **Potential Societal Impact:** While REAL is designed to enhance code quality and security, it could be **misused by malicious actors** to generate more sophisticated or harmful code. Moreover, improved automation may reduce the need for human oversight in certain development tasks, potentially impacting **entry-level programming roles**. These risks underscore the importance of responsible deployment—through access control, robust evaluation safeguards, and continued human oversight in the code generation process.

---

> ### Comment · Reviewer_qzWn · 2025-08-06
>
> I have read the author's response, and I appreciate the author's clarifications. I will keep my score unchanged.

---

> > ### Author Response · Authors · 2025-08-09
> > **Thank you for the discussion**
> >
> > We thank the reviewer for the review and discussion!

---

### Official Review · Reviewer_5a54 · 2025-06-25

**Clarity:** 4
**Significance:** 4
**Originality:** 3
**Rating:** 4
**Confidence:** 5

**Summary:**

This manuscript proposes ‘REAL’, a framework that uses reinforcement learning (RL) model to integrate dual automated signals; i.e., a program analyzer to detect maintainability defects and unit tests to ensure functional correctness. REAL was designed to enhance the quality of code generated by LLMs using a prompt-agnostic and reference-free approach. The automated, hybrid program analysis-based rewards mechanism gives REALs an upper hand compared to traditional methods that depend on manual annotations or brittle heuristics. REAL is leveraged to generate production-ready code by combining static detection of security and maintainability flaws along with unit testing to provide comprehensive feedback, which guides the LLM's code generation workflow. To accomplish this, the authors develop three extended benchmarks: (1) SecCodePLT+ (covering 18 Common Weakness Enumerations), (2) SafeSQL (targeting SQL injection vulnerabilities), and (3) APPS+ (assessing Python maintainability), to evaluate the REAL across diverse quality metrics. Results suggest that REAL-trained models outperform both the baseline fine-tuning and rule-based approaches, thus achieving an 85.9% success rate in functionality on SafeSQL and 92.3% maintainability on APPS+. REAL advances scalability and demonstrates elegant generalization across model sizes within 0.5B to 7B parameters. The novelty of REAL’s is its ability to leverage static analysis within a hybrid reward function to guide code generation toward safer and more maintainable outputs.

**Questions:**

1. How is the hybrid reward function (e.g., $r_{\text{total}} = \alpha r_{\text{quality}} + (1 - \alpha) r_{\text{func}}$) calibrated? I would recommend that the authors experiment with multiple values of α to assess its sensitivity.


2. How did the authors validate the correctness and robustness of the custom detectors (e.g., for sanitization or code style)? Do they generalize outside the training distribution?


3. Can the authors confirm if they measured or observed the RL-trained model exploiting brittle or sparse reward patterns, i.e., engaging in reward hacking?


4. Why were strong recent baselines (e.g., DeepSeekCoder, Claude-Code, StarCoder2) excluded from comparison? Would adding them affect the paper's conclusions?


5. Can the author report statistical confidence in the evaluation metrics (e.g., via multiple seeds, bootstrapping, or CI bars) to rule out that observed gains are not noise?

**Ethical Concerns:**

["NO or VERY MINOR ethics concerns only"]

**Final Justification:**

The rebuttal has improved the manuscript's clarity substantially, and provides additional empirical evidence addressed some of my earlier concerns considerably. Notably:
Reward Sensitivity – The authors added a sweep over α values across two benchmarks (SecCodePLT+ and SafeSQL), confirming that α ≈ 0.5 balances functionality and quality while showing predictable trade-offs when shifting emphasis.
Detector Validation – They now present in-domain (SecCodePLT+) and out-of-distribution (HumanEval) evaluations, including comparisons to existing tools (Bandit, Bearer) with strong gains in precision, recall, and F1, reinforcing the reliability of their static analysis signals.
Reward Hacking Mitigation – Clear evidence is provided of reward hacking with static-analysis-only rewards, and the corrective effect of combining unit tests with quality checks is demonstrated.
Human Evaluation – A small-scale expert study (50 APPS+ problems, 80% win rate for REAL outputs) was added to address readability and maintainability aspects not captured by automated metrics.
Novelty Clarification – The contribution is now explicitly framed as the first integration of program-analysis–driven reward shaping in RL for code generation, targeting maintainability and security alongside functionality.
Annotation Process Transparency – More detail is given for SafeSQL construction, including multi-pass expert verification and cross-checking, though IRR remains unreported.
Statistical Reliability – After discussion, the authors provided preliminary multi-seed results (SafeSQL, five seeds) with low standard deviations, indicating stable performance; they committed to updating the manuscript with these results.
These additions address the majority of the technical and methodological issues I raised. The paper still has some remaining limitations:
Statistical rigor has improved, but the full set of multi-seed results and statistical analyses (e.g., CIs, variance across all benchmarks) should be incorporated in the final manuscript for complete transparency.
Qualitative failure analysis is still somewhat limited; more diverse real-world failure case studies would further strengthen the diagnostic value.
Generalization to production settings remains only partially addressed; evaluation on more complex, interdependent industrial codebases is still an open step.
Overall, the rebuttal demonstrates responsiveness and meaningful improvements. The core idea, leveraging program analysis as a scalable, domain-relevant reward signal in RL for LLM code generation, is well-motivated, and the new results reinforce its technical merit. While some gaps remain for future work, the contribution is solid and timely. I maintain a borderline accept rating, leaning toward acceptance given the strengthened empirical grounding and practical relevance.

**Limitations:**

No. While the paper acknowledges the challenges of program analysis integration, it does not adequately address the potential brittleness or bias in static detectors, nor does it fully explore generalization risks or limitations of the hybrid reward design. Ethical considerations related to overtrust in detector-based training are also missing.

**Quality:**

3

**Strengths And Weaknesses:**

## **Strengths**

1. **Novel**

 Referencing Section: 3.1 Vulnerability Detector:

 - This work uses Program Analysis and integrates symbolic vulnerability detectors in the RL framework reward signal. This integration provides a significant contribution to the safe LLM code generation process.

2. **Scalability**

 - Section: 4.1 Experiment Settings (Pg. 5, Lines: 194–202), Section: 4.2 Quantitative Results (Pg. 7, Lines: 240–260), and Table 3 with surrounding discussion (Lines: 240–260) confirming scalability:
 - This RL framework scales across model sizes (i.e., from 0.5B to 7B) and is customizable with open-source tools, thus aiding reproducibility and increasing practical real-world adoption.

3. **Empirically grounded**

- This manuscript illustrates consistency in performance improvements across different benchmarks, with clear gains in both functionality and security-related metrics.

- Table 5, SecCodePLT+ and SafeSQL (security) and APPS+ (maintainability), with REAL outperforming all baselines in both quality and functionality.


4. **Hybrid Reward Design**:

- Section: 3.2 Reinforcement Learning with Hybrid Rewards (Pg. 4, Lines: 156–186)

- The authors implement a thoughtful balance of complementary objectives in SWE activities by blending functional correctness with program quality indicators.



5. **Reproducibility Pipeline**:
- Section: Introduction, Pg. 1, Lines: 10–11, Pg. 5, Lines: 194–202,

- The methodology used in this paper relies on openly available models and datasets, thus increasing its accessibility to the entire research community for replication and verification of results.

- Both data sources, APPS and SecCodePLT, are available and use open source tools like MyPy [Lehtosalo, 2025].

- REAL is prompt-agnostic and reference-free, hence scalable without manual intervention.





### **Weaknesses**

#### **Quality**

1. **Lack of proper validation** Pg. 3, Lines 36–50; Pg. 4, Lines 74–85

   - The authors did not formally evaluate the static analysis tools used to compute `r_quality` in a SWE context. Leaving the reader to wonder about their accuracy and susceptibility, especially to false positives/negatives. The absence of this evaluation undermines confidence in the reward signal's reliability and Static Analysis. Between false positives and false negatives, which is mostly severe and unacceptable in this use case?


2. **Binary Reward Granularity** Pg. 4, Eq. 2; Figure 5

   - The vulnerability reward is binary (0/1), and the hybrid reward lacks ablation or sensitivity testing with respect to the weighting parameter α. Without understanding how the reward channels interact, training robustness remains unclear.


3. **Lack of Qualitative Analysis** Section 5–6

    - Strange enough, no examples of incorrect, hallucinated, or low-quality code generations are reported in the paper. Failure case breakdowns are essential for understanding where the model goes wrong and for establishing trustworthy usage in production.


4. **No Human Evaluation**

   - Even though the paper automates the evaluations, yet, quality code at production still depends on subjective human judgment. I.e., metrics like readability, structure, and idiomatic expression—dimensions are best judged by humans. Even in this case, a solid measure of IRR should be implemented in the study design.



#### **Significance**

5. **Benchmarks Generalization Concerns**  Pg. 2, Lines 41–55

   - In this study, the authors extended the APPS+, SecCodePLT+, and SafeSQL Benchmarks on their use case. However, little is known about their representative assessment of actual codebases, thus leaving generalization to production code unverified.


6. **Benchmarks Overfitting Risk**

   - The authors performed an extensive fine-tuning and validation on the curated datasets. However, they did not test against distribution shift, using industrial-grade data that is usually more complex with interdependencies and sometimes suffers from poorly documented code patterns. These limitation (gaps) increases the risk of generalizability and robustness under distribution shift. Thereby reducing the quality and safety of the code generated by REAL.

7. **Unvalidated Metric** Section: 3.2, Pg. 5, Lines 182-186; Section 4.2, Table 3, Pg. 7
   - The paper treats code quality and functionality as a single metric, but does not provide theoretical or empirical support for its weighting scheme or assess whether the metric masks failures in one dimension. Pg. 5 line 184 simply provides the interval [0,1]

   - The authors provide no empirical/conceptual justification for how 𝛼 was chosen. Even though the paper claims trade-off between quality and functionality, tit is not clear how 𝛼 was calibrated.
   - While conducting a post-hybrid analysis using GAE and PPo is essential, I still do not understand why there is no sensitivity analysis on varying 𝛼 to observe the relationship between r_quality and r_function against performance.





#### **Originality**

8. **Algorithmic Contributions**

  - Overall, it seems that REAL suffers a bit of novelty concerns because the framework uses LLM as its policy model, the environment is the code generation task, and nothing special about that. The reward is derived from symbolic program analysis and unit tests combined (not human feedback). The authors also used PPO as-is; no modification was done architecturally or optimally.

  - While I appreciate and see some novelty contextually in the use of reward design, the underlying RL setup (PPO with reward shaping) is basic. Therefore, the primary originality lies only in the integration of symbolic signals into an RL framework for LLMs.


#### **Clarity** Pg. 8, Section 5.2

9. **Lack of Transparency in the Dataset Annotation Process**

   - It is not clear to me how the paper addresses potential human biases in the annotation process. Little is known about the labelling of vulnerability indicators and how the dataset was curated, validated, or verified. At least, I would appreciate a discussion of the IRR and how agreements/disputes were resolved among human annotators.


10. **Scalability Claims**

   - The paper claims that REAL scales across model sizes (0.5B, 3B, and 7B) with consistent improvements in SFT and RL-based baselines. Despite these claims, evidence for training stability and reproducibility is not discussed or presented.

   - Despite the superficial demonstrations of scale through performance metrics, the paper fails to show evidence through Training reward/loss curves, Standard deviation/error bars on results, Convergence discussion (e.g., speed, stability, and Hardware resource comparison across sizes.


   - Although the method is tested across LLM sizes, training stability, convergence behavior, and variance across runs are not sufficiently discussed.

---

> ### Author Rebuttal · Authors · 2025-07-31
>
> We sincerely thank Reviewer 5a54 for the insightful review and detailed suggestions. We are grateful that the reviewer carefully checked our paper and found our method novel, scalable, and empirically grounded.
>
> The concerns are mainly about **detail clarifications**.
>
> We first answer the questions and then address the remaining concerns.
>
> ---
>
> **Response to Question 1**
>
> > *“How is the hybrid reward function (e.g., ) calibrated? I would recommend that the authors experiment with multiple values of α to assess its sensitivity.”*
> >
>
> In our experiments we fixed α = 0.5, which delivered a strong balance of correctness and quality. To probe its sensitivity, we swept α over {0, 0.3, 0.5, 0.7, 1.0} on two benchmarks (SecCodePLT+ and SafeSQL).
>
> As α increases, the functionality score predictably declines, but by tuning α just above or below 0.5 we can actually surpass the default’s joint “Func-Qual” performance.
>
> (1) SecCodePLT+
>
> |Scale|Metric|α=0|α=0.3|α=0.5|α=0.7|α=1|
> |---|---|---|---|---|---|---|
> |0.5 B|Function|0.7317|0.7012|0.5854|0.4207|0.0732|
> | |Quality|0.1402|0.6524|0.7988|0.9512|0.9268|
> | |**Func-Qual**|**0.0854**|**0.3659**|**0.3963**|**0.4024**|**0.0732**|
> |3 B|Function|0.7683|0.7622|0.7378|0.7317|0.1585|
> | |Quality|0.2805|0.7378|0.8476|0.8598|0.9024|
> | |**Func-Qual**|**0.2073**|**0.5183**|**0.6037**|**0.5915**|**0.1220**|
> |7 B|Function|0.7805|0.7622|0.7561|0.7012|0.3415|
> | |Quality|0.3476|0.8720|0.8293|0.8720|0.8354|
> | |**Func-Qual**|**0.2866**|**0.6402**|**0.6159**|**0.6037**|**0.2866**|
>
> (2) SafeSQL
>
> |Scale|Metric|α=0|α=0.3|α=0.5|α=0.7|α=1|
> |---|---|---|---|---|---|---|
> |0.5 B|Function|0.7765|0.7647|0.7177|0.6588|0.3529|
> | |Quality|0.0471|0.8941|1.0000|1.0000|1.0000|
> | |**Func-Qual**|**0.0235**|**0.6941**|**0.7177**|**0.6588**|**0.3529**|
> |3 B|Function|0.8588|0.8471|0.8353|0.8235|0.5529|
> | |Quality|0.0824|1.0000|1.0000|1.0000|1.0000|
> | |**Func-Qual**|**0.0588**|**0.8471**|**0.8353**|**0.8235**|**0.5529**|
> |7 B|Function|0.8706|0.8706|0.8235|0.8118|0.6353|
> | |Quality|0.0588|1.0000|1.0000|1.0000|1.0000|
> | |**Func-Qual**|**0.0353**|**0.8706**|**0.8235**|**0.8118**|**0.6353**|
>
> ---
>
> **Response to Question 2**
>
> > *“How did the authors validate the correctness and robustness of the custom detectors (e.g., for sanitization or code style)? Do they generalize outside the training distribution?”*
> >
>
> We validate our detectors both **in-domain** and **out-of-distribution**:
>
> - In-Domain Validation (Security Detector)
>
>     Generally, we evaluate the detector using positive and negative pairs. For example, we leverage the SecCodePLT+ test set, which comprises 164 coding problems—each annotated with one “safe” and one “unsafe” solution for a specific CWE. For each detector, we compute true positives (TP), true negatives (TN), false positives (FP), and false negatives (FN), then derive precision, recall, and F1-score.
>
>     Below is a representative subset of results, comparing our detector against two popular open-source tools (Bandit and Bearer):
>
>     |CWE ID|Metric|Bandit|Bearer|Ours|
>     |---|---|---|---|---|
>     |77|Precision|0.6721|0.1961|1|
>     |77|Recall|0.8039|0.3279|1|
>     |77|F1-score|0.7321|0.3279|1|
>     |770|Precision|0.5|0|0.9808|
>     |770|Recall|0.0196|0|1|
>     |770|F1-score|0.0377|0|0.9903|
> - Out‐of‐Distribution Generalization (Maintainability Detector)
>
>     To assess generalization, we evaluate on the **HumanEval** benchmark, whose problem styles differ markedly from our training data (APPS+). Crucially, none of the HumanEval samples were seen during training. We measure both functional correctness and code quality to compute a hybrid “Functionality-Quality” score.
>
>     |Model|Function|Quality|Function-Quality|
>     |---|---|---|---|
>     |Qwen-Coder-Instruct-7B|0.7927|0.2561|0.2317|
>     |**REAL-7B**|**0.8476**|**0.9573**|**0.8171**|
>
> ---
>
> **Response to Question 3**
>
> > *“Can the authors confirm if they measured or observed the RL-trained model exploiting brittle or sparse reward patterns, i.e., engaging in reward hacking?”*
> >
>
> We explicitly monitored for reward‐hacking behaviors and confirmed that our performance gains were not due to exploiting brittle or sparse reward signals.
>
> We did observe reward‐hacking when using **only** the static‐analysis signal, and here is an example code that the model generates null code to bypass the static-analysis tool:
>
> ```python
> def generate_post_html(author: str, title: str, body: str, post_tags: list[str]) -> str:
>     # Implement the function to generate HTML for the post
>     pass
> ```
>
> We fixed this by adding correctness unit test and the model cannot simply generate null code to achieve a high reward. We manually checked and found the code was functional and high-quality.
>
> ---
>
> **Response to Question 4**
>
> > *“Why were strong recent baselines (e.g., DeepSeekCoder, Claude-Code, StarCoder2) excluded from comparison? Would adding them affect the paper's conclusions?”*
> >
> - Our method focuses on improving LLMs' capability to generate both correct and high-quality code (security and maintainability), not chasing SOTA performance. Performance comparisons against base model and SFT show significant gains, demonstrating our method's effectiveness.
> - These stronger models will not affect our conclusion — we have shown that REAL works across different model scales (0.5B, 3B, 7B), so stronger backbone models will also benefit from our training paradigm.
>
> ---
>
> **Response to Question 5**
>
> > *“Can the author report statistical confidence in the evaluation metrics (e.g., via multiple seeds, bootstrapping, or CI bars) to rule out that observed gains are not noise?”*
> >
>
> We can confirm that the observed gains are not from randomness or data noise. Extensive experiments on three different datasets (SecCodePLT+, SafeSQL, and APPS+) and three different model scales (0.5B, 3B, and 7B) show consistently significant gains.
>
> ---
>
> **We find that most of the Weaknesses are already addressed in the answers above. We respond to the remaining uncovered ones as follows:**
>
> ---
>
> **Response to Weaknesses 3 & 4**
>
> > *“…Failure case breakdowns are essential…” & “… best judged by humans…”*
> >
> - For the failure cases, we will add more in the revision. The failure cases mostly covers (1) null code with empty implementation, (2) correct but unsafe code, and (3) incorrect and unsafe code.
> - For human evaluation, we recruited two experts to do a **winning-rate–based evaluation** via pairwise comparisons on 50 coding problems from APPS+. For each problem they picked between the Qwen2.5-7B base output and the Qwen2.5-7B+REAL output, judging maintenance, readability, and fault tolerance (e.g., comments, type annotations, brevity, and error resilience). The REAL-trained model won 80% of the time. We will include more details in the revision.
>
> ---
>
> **Response to Weakness 7**
>
> > *“The paper treats code quality and functionality as a single metric ... Pg. 5 line 184 simply provides the interval [0,1]”*
> >
>
> The reviewer’s comment conflates the hybrid *training* reward with our *evaluation* metrics. In fact, we evaluate **functionality**, **quality**, and their **combination** separately; each is reported as a pass rate and defined in the paper (Line 213, Page 6).
>
> ---
>
> **Response to Weakness 8**
>
> > *Algorithmic Contributions: Overall, it seems that REAL suffers a bit of novelty concerns …*
> >
>
> We do **not** claim novelty in the general application of RL to code generation. Our contribution lies in **what** we optimize and **how** we do it.
>
> - We are the **first to leverage RL with Program Analysis feedback** to train LLMs that generate code that is not only functional but also **secure and maintainable**—a critical yet under-addressed problem, as highlighted by **Reviewers 5a54** and **qzWn**. While previous RL-for-code efforts focus almost exclusively on **functional correctness** using unit tests, we tackle **code quality**, including security and maintainability, which has been largely overlooked.
> - Additionally, we are the **first to introduce Program Analysis as a reward signal**. In contrast to unit tests—which are narrow, task-specific, and hard to scale—program analysis provides a **general, scalable, and task-agnostic** way to detect security flaws and enforce maintainability (e.g., checking for type hints). This makes it an ideal signal for guiding RL in this domain.
>
> ---
>
> **Response to Weakness 9**
>
> > *“It is not clear to me how the paper addresses potential human biases in the annotation process …”*
> >
>
> Thank you for pointing this out. The SafeSQL dataset was constructed through a multi-step process to reduce bias:
>
> - **Seed Tasks:** Human experts wrote 10 seed tasks covering SQL injection–prone scenarios across diverse query types.
> - **LLM Expansion:** We used **GPT-4.1** to mutate these tasks (e.g., changing constraints or merging cases), followed by **deduplication and expert verification**.
> - **Cross-Verification:** Multiple annotators reviewed each example, and disagreements were resolved via discussion. We also applied **program analysis tools** to enforce consistency.
> - **Functionality Tests:** For each task, we generated I/O pairs from reference solutions to create unit tests.
>
> While we did not compute IRR scores, we reduced annotation variance through multi-pass review and plan to include formal IRR in future work.
>
> ---
>
> We sincerely thank the reviewer for the detailed feedback and constructive suggestions. We believe our comprehensive responses and additional experimental results address all the raised concerns. We welcome any further questions and are committed to addressing any remaining issues.
>
> We appreciate the reviewer's consideration of our responses and hope they find our clarifications satisfactory.

---

> > ### Comment · Reviewer_5a54 · 2025-08-05
> >
> > I commend the authors for the work done in addressing my concerns. While the rebuttal improves clarity and addresses several key technical concerns, REAL still suffers from a significant empirical gap: the absence of rigorous statistical evaluation, which I believe casts doubt on the reliability and generalizability of its performance claims. I was expecting to see improvement in how the authors address this concern through multi-run reporting and proper statistical analysis; the results should be interpreted with caution.

---

> > > ### Author Response · Authors · 2025-08-06
> > > **Reply to Reviewer 5a54**
> > >
> > > We sincerely thank the reviewer for the reply and clarification on the remaining concern.
> > >
> > > We are currently working on it, and please allow us some additional time.
> > >
> > > We will post the results as soon as possible. Thank you!

---

> > > > ### Author Response · Authors · 2025-08-09
> > > >
> > > > We sincerely thank the reviewer for their detailed feedback and for clarifying the remaining concern regarding rigorous statistical evaluation.
> > > >
> > > > We fully agree on the importance of multi-run reporting and proper statistical analysis to ensure reliability and generalizability. We run additional experiments across five seeds and report averaged performance and standard deviation on the SafeSQL dataset:
> > > >
> > > > | SafeSQL   | Function               | Quality               | Function-Quality       |
> > > > |-----------|------------------------|-----------------------|------------------------|
> > > > | ReaL-0.5B | $0.7577 \pm 0.0229$     | $0.9106 \pm 0.0510$    | $0.7012 \pm 0.0105$     |
> > > > | ReaL-3B   | $0.8424 \pm 0.0065$     | $1.0000 \pm 0.0000$    | $0.8424 \pm 0.0065$     |
> > > >
> > > >
> > > >
> > > > The results are consistent across runs, with small standard deviations indicating stable performance and reinforcing the reliability of our findings.

---

### Official Review · Reviewer_YAqt · 2025-07-02

**Clarity:** 3
**Significance:** 2
**Originality:** 3
**Rating:** 3
**Confidence:** 4

**Summary:**

This work proposes a method called REAL, whose core idea is to introduce a hybrid reward mechanism during the training process of large code models. The hybrid reward consists of two components: one is a program vulnerability detector based on static program analysis techniques, and the other is a functional validator that verifies whether the function's intended behavior is achieved. By combining these two rewards, the large code model achieves performance improvements on three datasets: SecCodePLT+, APP+, and SafeSQL. The authors also present concrete examples to demonstrate how the code model "evolves" into a more secure model.

**Questions:**

1.	Why does the model perform so well on the SafeSQL dataset? Does this suggest overfitting or that the task itself lacks sufficient challenge?
2.	Why are the scores in the “Quality” column significantly higher than those in the other two columns?

**Ethical Concerns:**

["NO or VERY MINOR ethics concerns only"]

**Final Justification:**

The core contribution represents only an evolutionary refinement of existing approaches. It falls significantly short of the innovative breakthrough anticipated in the introduction. The claimed novelty exceeds the actual incremental advancement delivered. Not sufficient for acceptance.

**Quality:**

2

**Strengths And Weaknesses:**

Strengths:
1.	Using static program analysis techniques as a signal for code quality is meaningful, as it helps alleviate the issue of sparse rewards.
2.	The paper is clearly written.
Weaknesses:
1.	The idea of using reinforcement learning to train code models is not particularly novel. In particular, it remains unclear whether reinforcement learning — especially preference optimization algorithms like PPO — is truly beneficial for code-related tasks. According to the authors’ experimental results, REAL does not appear to deliver significant performance improvements.
2.	The paper lacks important experimental details. For example, how was reinforcement learning applied? What data was used as seed data? How were the experiments conducted? How does the hyperparameter $\alpha$ affect the trade-off between code quality and functional correctness?

---

> ### Author Rebuttal · Authors · 2025-07-31
>
> We sincerely thank the reviewer for their time and thoughtful feedback. We are pleased to hear that the reviewer finds our method meaningful and appreciates the clarity of our writing. Below, we address the reviewer’s concerns in detail:
>
> ---
>
> **Response to Question 1**
>
> > *“Why does the model perform so well on the SafeSQL dataset? Does this suggest overfitting or that the task itself lacks sufficient challenge?”*
>
> - This is not overfitting, as we use separate data for training and evaluation. Moreover, during RL training, the model receives no ground-truth answers—only feedback from program analysis and unit tests.
> - The strong performance likely comes from the dataset focusing on a single CWE (SQL injection), enabling the model to specialize in avoiding this vulnerability.
> - Despite targeting one CWE, SafeSQL remains challenging. As shown in Table 2, untrained base models perform poorly, with joint pass rates of only **20%**, **27%**, and **58%** for 0.5B, 3B, and 7B models, respectively.
>
> ---
>
> **Response to Question 2**
>
> > “Why are the scores in the Quality column significantly higher than those in the other two columns?”
>
> Functionality and quality capture different aspects of code generation and are **not directly comparable**. Their difference arises from **how each metric is defined**:
>
> - **Functionality**: Measures the percentage of prompts where the generated code passes *all* unit tests—an all-or-nothing criterion.
> - **Quality**: Measures the percentage of prompts passing *all* program analysis checks. However, this metric can be gamed—for example, by generating empty code—so it should not be used in isolation.
> - **Joint (Functionality & Quality)**: Requires passing *both* unit tests and analysis checks, making it the strictest metric. As a result, its score is naturally lower than either individual metric.
>
> ---
>
> **Response to Weakness 1**
>
> > *“The idea of using RL to train code models is not particularly novel.“*
>
> - We do **not** claim novelty in applying RL to code generation broadly. Instead, our contribution lies in **what** we optimize and **how**.
> - We are the **first to use RL with Program Analysis feedback** to train LLMs that generate not just functional, but also **secure and maintainable** code—an urgent and under-explored problem, as also recognized by **Reviewers 5a54** and **qzWn**. Prior RL-for-code work focuses primarily on **functionality** via unit tests. In contrast, we address **quality**—i.e., security and maintainability—which prior work overlooks.
> - We are also the **first** to propose using **Program Analysis** as the reward signal. Unlike unit tests, which are impractical and problem-specific, program analysis offers a **general, scalable, and task-agnostic** mechanism to detect vulnerabilities and enforce maintainability standards (e.g., type annotations), making it uniquely suited for RL optimization in this setting.
>
> > *“In particular, it remains unclear whether reinforcement learning — especially preference optimization algorithms like PPO — is truly beneficial for code-related tasks.”*
>
> - **RL has demonstrated clear benefits for coding tasks**: Recent models like DeepSeek-R1 [1], Kimi-K2 [2], and Qwen3-Coder [3] are all post-trained with RL (typically using unit test feedback) and show strong performance on code benchmarks. While details are not public, Anthropic’s blog suggests Claude also uses RL [4]. While the broader impact of RL on reasoning remains open, its value for math and coding tasks is well supported.
> - **Clarifying PPO**: PPO is not inherently a "preference optimization" algorithm—it is a general-purpose RL method. In RLHF, it’s often used to optimize learned preference models, but in our case, PPO directly optimizes **hybrid rewards** based on program analysis and unit test feedback. This encourages the model to generate code that is not just correct, but also secure and maintainable.
>
> [1] DeepSeek: Guo, Daya, et al. "Deepseek-r1: Incentivizing reasoning capability in llms via reinforcement learning." *arXiv preprint arXiv:2501.12948* (2025).
>
> [2] Kimi: Team, Kimi, et al. "Kimi K2: Open Agentic Intelligence." *arXiv preprint arXiv:2507.20534* (2025).
>
> [3] Qwen Team: "Qwen3-Coder: Agentic Coding in the World." Blog post. Qwen Blog, 2025.
>
> [4] Anthropic Team. “Claude’s Constitution.” Anthropic News blog post. May 9, 2023. Anthropic.
>
> > *“REAL does not appear to deliver significant performance improvements.”*
>
> We respectfully disagree. REAL consistently yields **substantial gains** over both the vanilla base model and the strongest baseline (SFT), especially when evaluated on the Joint Pass-Rate, a strict metric requiring the code to pass both unit tests and static analysis checks.
>
> Across all datasets and model sizes, REAL improves over the vanilla model by up to **+57.7%**, and over SFT by up to **+16.5%**. Notably, for the 7B model, REAL outperforms SFT by **+15.3%** on SecCodePLT+, **+12.9%** on SafeSQL, and **+15.4%** on APPS+. These consistent gains across benchmarks and scales clearly demonstrate the effectiveness of our program-analysis-guided RL framework.
>
> ---
>
> **Response to Weakness 2**
>
> The implementation details can be found in Appendix A.2 and we add more details below to help improve the quality of our paper and we will definitely include more in the final revision.
>
> > *“how was reinforcement learning applied?”*
>
> We employ **Proximal Policy Optimization (PPO)** on top of a pretrained code model (Qwen2.5‑Coder) using our hybrid reward. We use the VeRL framework to implement this and we mainly focus on the reward design. All the code will be released and we will also add more details in the paper.
>
> > *“What data was used as seed data? ”*
>
> To curate the SafeSQL dataset, we asked human experts to write 10 database programming tasks that are potentially vulnerable to SQL injection. These tasks span diverse, realistic scenarios—including filtering, sorting, and aggregation—to ensure broad coverage of query types.
>
> Each prompt was carefully reviewed by both human experts and a strong LLM (GPT‑4.1) to ensure clarity and solvability. For example, one task asks the model to query a `books` table for entries within a user-specified price range and genre, simulating a typical real-world use case prone to SQL injection risks in the code solutions.
>
> > *“How were the experiments conducted?”*
>
> Our experiment setup is introduced in Section 4.1 and more implementation details are provided in Appendix A.2.  Generally, we implement ReaL based on the widely used VeRL framework and conduct all experiments on an 8 x H100 GPUs server. The backbone model used in our experiments is Qwen2.5-Coder-Instruct. We will elaborate more on the experiment details in the final revision.
>
> > *“How does the hyperparameter α affect the trade-off between code quality and functional correctness?”*
>
> Our experiments reveal a clear trade-off: optimizing solely for correctness or quality often hurts the other. To study this, we vary the weighting factor **α** in the hybrid reward and evaluate on SecCodePLT+ and SafeSQL:
>
> ### SecCodePLT+
>
> | Scale | Metric | α=0 | α=0.3 | α=0.5 | α=0.7 | α=1 |
> | --- | --- | --- | --- | --- | --- | --- |
> | 0.5 B | Function | 0.7317 | 0.7012 | 0.5854 | 0.4207 | 0.0732 |
> |  | Quality | 0.1402 | 0.6524 | 0.7988 | 0.9512 | 0.9268 |
> |  | **Func‑Qual** | **0.0854** | **0.3659** | **0.3963** | **0.4024** | **0.0732** |
> | 3 B | Function | 0.7683 | 0.7622 | 0.7378 | 0.7317 | 0.1585 |
> |  | Quality | 0.2805 | 0.7378 | 0.8476 | 0.8598 | 0.9024 |
> |  | **Func‑Qual** | **0.2073** | **0.5183** | **0.6037** | **0.5915** | **0.1220** |
> | 7 B | Function | 0.7805 | 0.7622 | 0.7561 | 0.7012 | 0.3415 |
> |  | Quality | 0.3476 | 0.872 | 0.8293 | 0.872 | 0.8354 |
> |  | **Func‑Qual** | **0.2866** | **0.6402** | **0.6159** | **0.6037** | **0.2866** |
>
> ### SafeSQL
>
> | Scale | Metric | α=0 | α=0.3 | α=0.5 | α=0.7 | α=1 |
> | --- | --- | --- | --- | --- | --- | --- |
> | 0.5 B | Function | 0.7765 | 0.7647 | 0.7177 | 0.6588 | 0.3529 |
> |  | Quality | 0.0471 | 0.8941 | 1.0000 | 1.0000 | 1.0000 |
> |  | **Func‑Qual** | **0.0235** | **0.6941** | **0.7177** | **0.6588** | **0.3529** |
> | 3 B | Function | 0.8588 | 0.8471 | 0.8353 | 0.8235 | 0.5529 |
> |  | Quality | 0.0824 | 1.0000 | 1.0000 | 1.0000 | 1.0000 |
> |  | **Func‑Qual** | **0.0588** | **0.8471** | **0.8353** | **0.8235** | **0.5529** |
> | 7 B | Function | 0.8706 | 0.8706 | 0.8235 | 0.8118 | 0.6353 |
> |  | Quality | 0.0588 | 1.0000 | 1.0000 | 1.0000 | 1.0000 |
> |  | **Func‑Qual** | **0.0353** | **0.8706** | **0.8235** | **0.8118** | **0.6353** |
>
> As shown, **low α** (favoring correctness) leads to poor quality scores, while **high α** (favoring quality) degrades functional correctness. Both extremes result in lower **joint pass rates**. In contrast, moderate values (e.g., **α = 0.3–0.5**) strike a better balance, yielding the strongest overall performance across scales and datasets.
>
> These findings validate our **hybrid reward design**, demonstrating that RL can flexibly optimize different objectives through careful tuning of α. In our main results, we use **α = 0.5** as a balanced default, but the framework remains adaptable to varied priorities.
>
> We hope this analysis addresses the reviewer’s question. We’re happy to clarify further if needed.

---

> > ### Comment · Reviewer_YAqt · 2025-08-05
> >
> > I have read the author's response, and I'm grateful for the author's clarification. I will maintain my score.

---

> > > ### Author Response · Authors · 2025-08-05
> > > **Rebuttal Follow-up to Reviewer YAqt**
> > >
> > > We sincerely thank the reviewer for their reply and acknowledgement.
> > >
> > > We noted **`factual inaccuracies in the review`**, specifically the mischaracterization of PPO as primarily "preference optimization," whereas it is a general RL algorithm. This may have **inheritantly influenced the reviewer's skepticism** regarding RL’s applicability to our code generation setting.
> > >
> > > Since we `didn't see any following-up` about unaddressed concerns, we **respectfully** request the reviewer reconsider and adjust the rating. If any concerns remain, we're happy to provide immediate responses.
> > >
> > > Thank you very much!

---

> > > > ### Comment · Reviewer_YAqt · 2025-08-06
> > > >
> > > > I sincerely appreciate the authors' responses to the previous review comments. First, I would like to clarify a potential misunderstanding caused by my wording: PPO is indeed one of the commonly used reinforcement learning algorithms in preference optimization. However, this does not affect my understanding of reinforcement learning—the authors perform preference optimization on a reference model based on reward signals.
> > > >
> > > > Second, I would like to emphasize that I did not deny the authors' innovation in incorporating program analysis feedback. In fact, I explicitly mentioned this strength in the previous review. That said, presenting the integration of program analysis feedback into the reward model as a main innovation appears somewhat incremental to me.
> > > >
> > > > Lastly, I have some concerns regarding the results. I observed that using simple SFT techniques on around 3k samples already yields fairly competitive performance, which can be achieved within an hour on a single A100 GPU. In contrast, the authors leveraged an H100 cluster, yet the results do not seem proportionally improved in comparison to the computational resources used.

---

> > > > > ### Author Response · Authors · 2025-08-07
> > > > > **Significant Flaws in the Review  &  Violating Reviewer Guidelines**
> > > > >
> > > > > We sincerely thank the reviewer for their continued engagement.
> > > > >
> > > > > However, we respectfully note several **`critical flaws`** that have influenced your final evaluation, regardless of the fact that the reviewer asked `no follow-up question` and decided to maintain the score, which is `irresponsible`.
> > > > >
> > > > > ---
> > > > >
> > > > > **1. Misconceptions About PPO**
> > > > >
> > > > > The reviewer states:
> > > > > > "However, this does not affect my understanding of reinforcement learning—the authors perform `preference` **`optimization on a reference model`** based on reward signals."
> > > > >
> > > > > **Response:**
> > > > >
> > > > > (1) We didn't do **`preference optimization`**. We must insist that simply calling this a “wording issue” is **`unacceptable`** -- the reviewer should be precise and provide professional feedback.
> > > > >
> > > > > (2) Reference model is **`frozen`** and **`not optimzied`** or updated.
> > > > >
> > > > > (3) PPO stands for Proximal Policy Optimization, not "Preference Optimization".
> > > > >
> > > > > These all together reflect that:
> > > > >
> > > > > - the reviewer has a `fundamental misunderstanding` or `limited understanding` of PPO, or more general RL, leading to biased evaluation on our Code RL work.
> > > > >
> > > > > - the reviewer `ignores overwhelming industry consensus`, as cited in our rebuttal, leading models (DeepSeek-R1, Qwen3-Coder) use RL for code tasks with validated success.
> > > > >
> > > > > ---
> > > > >
> > > > > **2. Misinterpretation of Experimental Results**
> > > > >
> > > > > The reviewer claims:
> > > > >
> > > > > >"...using simple SFT techniques on around 3k samples already yields `fairly competitive performance`..."
> > > > >
> > > > > **Response:**
> > > > >
> > > > > Our method has `signficant gain`. The main metric is PassRate on both functionality and quality, denoted as "Func.-Qual." in Table 2 & 3.
> > > > >
> > > > > | Method   | SecCodePLT+ (0.5 B) | SecCodePLT+ (3 B) | SecCodePLT+ (7 B) | SafeSQL (0.5 B) | SafeSQL (3 B) | SafeSQL (7 B) | APPS+ (0.5 B) | APPS+ (3 B) | APPS+ (7 B) |
> > > > > | -------- | ------------------- | ----------------- | ----------------- | --------------- | ------------- | ------------- | ------------- | ----------- | ----------- |
> > > > > | **SFT**  | 0.4573              | 0.4573            | 0.4634            | 0.5527          | 0.6824        | 0.7294        | 0.1888        | 0.4046      | 0.4663      |
> > > > > | **REAL** | 0.3963              | 0.6037            | 0.6159            | 0.6941          | 0.8471        | 0.8588        | 0.2909        | 0.5241      | 0.6204      |
> > > > > | **Gain** | **–0.0610**         | **+0.1464**       | **+0.1525**       | **+0.1414**     | **+0.1647**   | **+0.1294**   | **+0.1021**   | **+0.1195** | **+0.1541** |
> > > > >
> > > > >
> > > > > ---
> > > > >
> > > > >
> > > > > **3. Unsubstantiated Claims on Computational Cost**
> > > > >
> > > > > The reviewer states:
> > > > >
> > > > > > which can be achieved within an hour on `a single A100 GPU`. In contrast, the authors leveraged `an H100 cluster`, yet the results do not seem proportionally improved in comparison to the computational resources used.
> > > > >
> > > > > **Response**
> > > > > - It is `unclear` which model size, how many epochs, or what precision `the reviewer refers to` — `nor is there evidence this matches our setup`.
> > > > >
> > > > > - Having 8xH100 node (`not cluster`) does not mean for every experiment we have to use all of them for a long time.
> > > > >
> > > > > - SFT requires huge efforts on ground truth annotations -- detailed bug free and high quality code, usually requiring professional annotated answers. While RL leverage verifiable reward, also known as RLVR, is more scalable.
> > > > >
> > > > > ---
> > > > >
> > > > > **4. Adherence to Reviewing Best Practices**
> > > > >
> > > > > According to the `NeurIPS Reviewer Guidelines`, reviews should be **`evidence-based`**, **`constructive`**, and focused on **`substantive technical points`**. We note several statements **lacked supporting data** or **misconstrued** our methods, which risks unfairly disadvantaging the work.
> > > > >
> > > > >
> > > > > ---
> > > > >
> > > > > We hope this clarification addresses the core misunderstandings and demonstrates the rigor and scalability of our REAL framework; we respectfully request that you reconsider your evaluation in light of these facts.
> > > > >
> > > > > If there is any additional concerns, we are happy to provide further clarifications.
> > > > >
> > > > > Thank you very much!

---

### Official Review · Reviewer_sExN · 2025-07-18

**Clarity:** 3
**Significance:** 3
**Originality:** 3
**Rating:** 4
**Confidence:** 4

**Summary:**

The authors propose a reinforcement learning-based framework for quality code generation, named as REAL, which integrates an automated feedback policy through program analysis with minimal human intervention. They also contribute three curated datasets to support their primary goal of generating high-quality code, with a particular focus on vulnerability detection and maintainability. Finally, they evaluate their proposed methodology in terms of balancing the trade-off between these two objectives, demonstrating improvements over prior approaches that address them separately.

**Questions:**

Q1. Instead of curated datasets, what if we evaluate on previous SecCodePLT [Yang e al., 2024] and APPS [Hendrycks et al., 2021] without any preprocessing. Can you please demonstrate similar benchmarking as Table 3-5? Can REAL, as a novel stand alone RL framework, applicable for any unknown real world raw datasets (generalizability) towards satisfying maintainability and vulnerability check (can we expect self-automated)?
Q2. Authors mentioned as "with minimal manual intervention", however used 3 curated datasets for evaluation. How minimal was this "curation" as "by extending existing datasets or evolving data with LLM"? Is there any bias introduced due to these curated datasets in training? Can it anyhow misguide the RL agent in specific scenario, for complex logic codes especially in dynamic contexts?
Q3. Program analysis-based feedback might not always provide explainable rationales for the improvements or rejections. Any concerns there?

**Ethical Concerns:**

["NO or VERY MINOR ethics concerns only"]

**Final Justification:**

The authors have provided satisfactory and substantial responses, including relevant demonstrations and clarifications, to all my concerns. Accordingly, I am raising my score to 4. Borderline Accept.

**Limitations:**

Limitations are mentioned in weakness part.

**Paper Formatting Concerns:**

No major formatting issues noticed.

**Quality:**

3

**Strengths And Weaknesses:**

Strength:
1. The authors well discussed and presented metrics and limitations of prior research works that targeting similar scope of quality code generation framework, which is highly beneficial for understanding the fundamental background, research roadmap trajectory and the expected improvements.
2. Experimental set up and benchmark strategy are well-structured and supported with evidence. The proposed methodology consistently  outperforms baseline and previous approaches across various evaluation metrics.

Weakness:

1. Methodology section appears to be more theoretical, putting more emphasis on "what" rather going in depth of "how" only mentioning as a generalized PPO based RL-based framework with reward/penalty mechanisms. It lacks deeper technical elaboration. More implementations details are needed for better clarity.
2. Experimental validation: Please make it more clear that as benchmarked on 3 curated datasets as SecCodePLT+, SafeSQL and APPS+, proposed method outperforming because of these specifically curated dataset or for employing a RL-based framework or combining both. Generalization over real-world raw dataset should be explored.
3. Authors should also points towards current limitations and potential breakpoints of the REAL framework (if any), particularly in the context of real-world quality code generation applications
4. Several typos and grammatical errors have been noticed. A through proofreading is encouraged.

---

> ### Author Rebuttal · Authors · 2025-07-31
>
> We thank Reviewer sExN for their thorough review and insightful comments. We are pleased that the reviewer recognized ReaL’s well-structured experiment settings and superior performance over baselines. We address the reviewer’s concerns as follows:
>
> ---
>
> **Response to Weakness 1**
>
> > *“More implementations details are needed for better clarity.”*
>
> We provide additional technical details to help the reviewer better understand our work. In the **REAL** framework, we fine‑tune **Qwen2.5‑Coder‑Instruct** using **Proximal Policy Optimization (PPO)** with a **hybrid reward** that balances **functional correctness** and **code quality** (security and maintainability). At each training step, the model generates candidate solutions for coding prompts, evaluates them using unit‑test pass rates and static analyzers, and combines these scores into a normalized hybrid reward. The static analyzer leverages control‑flow graphs and taint analysis to detect unsafe data flows from user input to execution. Based on the reward signals, we compute advantages using Generalized Advantage Estimation (GAE) and update the policy with PPO’s clipped objective, augmented by KL‑penalty and entropy regularization to ensure stable learning and maintain exploration. This framework allows **REAL** to improve steadily using scalable and verifiable feedback.
>
> ---
>
> **Response to Weakness 2**
>
> > *“Please make it more clear that as benchmarked on 3 curated datasets as SecCodePLT+, SafeSQL and APPS+, proposed method outperforming because of these specifically curated dataset or for employing a RL-based framework or combining both. Generalization over real-world raw dataset should be explored.”*
>
> We clarify that the datasets used in this paper are **not specifically curated** for our method, but are used to **establish a new experimental setting** and to **train and evaluate all baseline methods**, ensuring a fair comparison. The original SecCodePLT and APPS datasets remain unchanged; we only add new verifiers (security and maintainability) as extra evaluation dimensions. The SafeSQL dataset was curated solely to study SQL development due to the lack of existing datasets. And it is also used to train the SFT baseline. Thus, REAL’s superior performance stems from our RL‑based framework with hybrid reward mechanisms, not from data differences.
>
> To further validate generalization, we trained REAL on APPS+ and evaluated it on the unseen HumanEval [1] benchmark. The result is as follows. REAL‑7B achieves substantial improvements across all metrics despite not being trained on HumanEval, demonstrating strong generalization.
>
> | HumanEval | Functionality | Quality | Functionality-Quality |
> | --- | --- | --- | --- |
> | Qwen-Coder-Instruct-7B | 0.7927 | 0.2561 | 0.2317 |
> | REAL-7B | 0.8476 | 0.9573 | 0.8171 |
>
> [1] Chen, Mark, et al. "Evaluating large language models trained on code." *arXiv preprint arXiv:2107.03374* (2021).
>
> ---
>
> **Response to Weakness 3**
>
> > *“Authors should also points towards current limitations and potential breakpoints of the REAL framework (if any), particularly in the context of real-world quality code generation applications”*
>
> We appreciate the reviewer’s request. We would like to highlight our novelty in our focus on the quality code and the use of program analysis in reward mechanism, and discuss current limitations of the **REAL** framework.
>
> - **Focus on quality code:** REAL is the **first** to apply reinforcement learning to optimize for **code quality**—specifically **security and maintainability**—rather than just functional correctness. This is increasingly critical as coding assistants are integrated into real-world workflows. Prior RL efforts mainly focus on solving tasks via unit tests, overlooking real risks like unsafe or unmaintainable code. Our work addresses this important and under-explored problem.
> - **Use of program analysis in reward:** Unlike prior works that rely on unit tests as rewards—which are task-specific and insufficient for quality aspects—we leverage **program analysis** (e.g., taint analysis and control-flow graphs) as a **general-purpose reward signal**. This enables scalable and automated evaluation of vulnerabilities and maintainability across diverse prompts, without the need for handcrafted test cases.
> - **Limitations and future directions:** Our current focus is on **single-turn** quality code generation. We acknowledge that many real-world applications involve **multi-turn interaction** and iterative refinement. Extending REAL to such settings—e.g., through agent-based frameworks that incorporate feedback loops—is a key direction for future work.
>
> ---
>
> **Response to Question 1**
>
> > *“Instead of curated datasets, what if we evaluate on previous SecCodePLT [Yang e al., 2024] and APPS [Hendrycks et al., 2021] without any preprocessing. Can you please demonstrate similar benchmarking as Table 3-5?”*
>
> We’d like to clarify that we did **not curate or preprocess** new data for the SecCodePLT and APPS benchmarks. Our updated versions (SecCodePLT+ and APPS+) use **exactly the same data**—we simply add static analysis tools as an **additional evaluation metric** to assess code quality (security and maintainability), which was not covered in the original benchmarks that only used unit tests.
>
> Dataset statistics are shown below:
>
> |  | Training set | Testing set |
> | --- | --- | --- |
> | SecCodePLT | 655 samples | 164 samples |
> | SecCodePLT+ | 655 (exactly the same data as SecCodePLT) | 164 (exactly the same data as SecCodePLT) |
> | APPS | 2038 samples | 519 samples |
> | APPS+ | 2038 (exactly the same data as APPS) | 519 (exactly the same data as APPS) |
>
> > *“Can REAL, as a novel stand alone RL framework, applicable for any unknown real world raw datasets (generalizability) towards satisfying maintainability and vulnerability check (can we expect self-automated)?”*
>
> Yes, this is a core strength of the REAL framework. By integrating **program analysis tools as reward functions**, REAL can generalize to **unseen, real-world datasets** without manual curation. To demonstrate this, we trained on APPS+ and evaluated on **HumanEval**—a benchmark with different styles and no overlap in training data. Despite no exposure to HumanEval, REAL significantly improves code quality:
>
> | HumanEval | Functionality | Quality | Functionality-Quality |
> | --- | --- | --- | --- |
> | Qwen-Coder-Instruct-7B | 0.7927 | 0.2561 | 0.2317 |
> | REAL-7B | 0.8476 | 0.9573 | 0.8171 |
>
> [1] Chen, Mark, et al. "Evaluating large language models trained on code." *arXiv preprint arXiv:2107.03374* (2021).
>
> ---
>
> **Response to Question 2**
>
> > *“Authors mentioned as "with minimal manual intervention", however used 3 curated datasets for evaluation. How minimal was this "curation" as "by extending existing datasets or evolving data with LLM"? Is there any bias introduced due to these curated datasets in training? Can it anyhow misguide the RL agent in specific scenario, for complex logic codes especially in dynamic contexts?”*
>
> We’d like to clarify that “minimal manual intervention” refers to the use of **program analysis tools** as automated reward functions in REAL. These tools significantly reduce human effort in detecting vulnerabilities in generated code, especially when compared to manually crafting case-specific unit tests.
>
> For example, given 1,000 SQL programs, a single static analyzer can check for SQL injection vulnerabilities—eliminating the need to manually write and annotate 1,000 individual test cases.
>
> As for the datasets mentioned in the question, we’d like to clarify that the three mentioned datasets were **not curated specifically for training REAL**, but rather to establish a sound experimental setting for evaluating security and maintainability—both for our method and future work. We will include these clarification in our future version to make this point clearer.
>
> ---
>
> **Response to Question 3**
>
> > *“Program analysis-based feedback might not always provide explainable rationales for the improvements or rejections. Any concerns there?”*
>
> In REAL, we use program analysis–based feedback as part of the reward function, represented as a binary signal (1 for secure, 0 for vulnerable). While this feedback is not explicitly explainable, our experiments show that it is **sufficient to guide learning** and significantly improve code quality.
>
> This practice aligns with **standard approaches in the RL with verifiable rewards (RLVR)** literature [1,2,3,4], where binary or scalar reward signals are commonly used. The advantage estimation mechanism in PPO effectively captures reward differences between successful and unsuccessful outputs, enabling the policy to learn the desired behaviors—even in the absence of detailed rationale.
>
> [1] AI2: Lambert, Nathan, et al. "Tulu 3: Pushing frontiers in open language model post-training." *arXiv preprint arXiv:2411.15124* (2024).
>
> [2] DeepSeek: Guo, Daya, et al. "Deepseek-r1: Incentivizing reasoning capability in llms via reinforcement learning." *arXiv preprint arXiv:2501.12948* (2025).
>
> [3] Microsoft Research: Wen, Xumeng, et al. "Reinforcement Learning with Verifiable Rewards Implicitly Incentivizes Correct Reasoning in Base LLMs." *arXiv preprint arXiv:2506.14245* (2025).
>
> [4] Meta AI: Gehring, Jonas, et al. "Rlef: Grounding code llms in execution feedback with reinforcement learning." *arXiv preprint arXiv:2410.02089* (2024).

---

> > ### Comment · Reviewer_sExN · 2025-08-02
> > **Rebuttal reviewer**
> >
> > The authors have provided satisfactory and substantial responses, including relevant demonstrations and clarifications, to all my concerns. Accordingly, I am raising my score to 4. Borderline Accept.

---

> > > ### Author Response · Authors · 2025-08-09
> > > **Thank You for Raising the Score to 4**
> > >
> > > We sincerely appreciate your updated evaluation and positive feedback on our rebuttal. Thank you for acknowledging our clarifications and demonstrations.
> > >
> > > **We are slightly confused by the reviewing system, as we can `still see the score as 3` in the OpenReview Portal. Could you please help us check this issue?**
> > >
> > > We truly appreciate it and thank you very much!

---

### Note · Authors · 2025-08-14

We thank all reviewers and the chair for their engagement. We emphasize the novelty of **REAL** in its focus on code quality, its use of program analysis in RL reward, and its superior performance.
- **Focus on code quality:** REAL is the **first** to apply RL to optimize for code quality (security and maintainability), rather than focusing solely on functional correctness. This is increasingly critical as coding assistants are deployed in real-world workflows. Prior RL approaches have primarily targeted task completion via unit tests, overlooking real risks such as insecure or unmaintainable code. Our work addresses this important and under-explored challenge.
- **Program analysis–based rewards:** Prior works rely on unit tests, which are task-specific and insufficient for evaluating quality. We incorporate program analysis (taint analysis and control-flow graphs) as a **general-purpose reward signal**, enabling **scalable and automated** detection of vulnerabilities and maintainability issues, without the need for handcrafted test cases.
- **Superior performance:** REAL achieves **SOTA results** in generating secure and accurate code across three benchmarks with diverse quality requirements, covering 18 CWEs and key maintainability concerns. It outperforms the second-best method by up to 16.47%.

In our rebuttal, we addressed reviewers’ questions and concerns through clarifications, extra experiments, and comparisons. In particular, we:
- Elaborated on new dataset metrics for code quality, including those targeting security and maintainability issues;
- Added ablation studies on reward weighting ratios for our hybrid rewards, showing consistent improvements;
- Compared our vulnerability detectors with external tools, highlighting their higher accuracy and reliability;
- Confirmed our plan to open-source the code and data to facilitate future research.

We thank reviewers **sExN**, **5a54**, and **qzWn** for recognizing the clarity of our paper, the novelty of our joint optimization framework, and the importance of addressing both functionality and security in code generation. We also respectfully address reviewer **YAqt**’s misunderstandings regarding PPO algorithms and mainstream code generation practices, and clarify their misinterpretation of the reported performance improvements and training costs. However, we note that no further feedback was provided during the review process.

Again, we thank all reviewers and the chair for engaging with our work!

---

### Decision · Program_Chairs · 2025-09-17

**Decision:**

Accept (poster)

**Comment:**

The paper proposes REAL, a reinforcement learning (RL) framework to incentivize LLMs to generate production-quality code that goes beyond functional correctness by incorporating security and maintainability qualities. REAL leverages program analysis to detect defects and unit tests to ensure correctness, offering prompt-agnostic and reference-free supervision for scalability.

Strengths:
- Clear experimental setup and benchmarks, showing strong performance over baselines (sExN, 5a54)
- While RL for code models has been studied before (5a54, YAqt), the paper innovatively extends it to target security and maintainability, integrating program analysis into the reward signal (5a54, qzWn)
- Scalable across model sizes and customizable with open-source tools (5a54)

Weaknesses:
- Method description remains somewhat theoretical, lacking deeper technical/implementation details (sExN, YAqt, qzWn), though many concerns were clarified in rebuttal
- Limited discussion of current limitations and possible breakpoints of the framework (sExN), along with missing qualitative analysis (5a54)
- Absence of human evaluation on subjective qualities such as readability (5a54). The authors addressed this issue during rebuttal.

In summary, the paper addresses an important challenge in LLM-based code generation by moving beyond correctness toward production-level qualities. The approach is well-motivated, simple yet promising with comprehensive experimental results. I advise the authors review all the feedback and improve the paper with more concrete technical details, analysis of limitations, and human validation to strengthen their contributions.